# GHOST ON THE SHELL

## AN EXPRESSIVE REPRESENTATION OF GENERAL 3D SHAPES

**Zhen Liu[1,2]  Yao Feng[1,3,*]  Yuliang Xiu[1,*]  Weiyang Liu[1,4,*]  Liam Paull[2]  Michael J. Black[1,‡]  Bernhard Schölkopf[1,‡]**

[1]Max Planck Institute for Intelligent Systems - Tübingen   [2]Mila – Quebec AI Institute, Université de Montréal
[3]ETH Zürich   [4]University of Cambridge   [*]Equal contribution   [‡]Shared last author

Project page: `gshell3d.github.io`

(a) **Ghost-on-the-Shell**: A general 3D representation

(b) **Application I**: Reconstruction from multi-view images

(c) **Application II**: Unconditional mesh generation

Figure 1: Left: Illustration of mesh extraction with **G-SHELL** through a manifold signed distance on a surface; Right: Applications of **G-SHELL**, multiview mesh reconstruction (top) and mesh generation (bottom).

## ABSTRACT

The creation of photorealistic virtual worlds requires the accurate modeling of 3D surface geometry for a wide range of objects. For this, meshes are appealing since they 1) enable fast physics-based rendering with realistic material and lighting, 2) support physical simulation, and 3) are memory-efficient for modern graphics pipelines. Recent work on reconstructing and statistically modeling 3D shape, however, has critiqued meshes as being topologically inflexible. To capture a wide range of object shapes, any 3D representation must be able to model solid, watertight, shapes as well as thin, open, surfaces. Recent work has focused on the former, and methods for reconstructing open surfaces do not support fast reconstruction with material and lighting or unconditional generative modelling. Inspired by the observation that open surfaces can be seen as islands floating on watertight surfaces, we parameterize open surfaces by defining a manifold signed distance field on watertight templates. With this parameterization, we further develop a grid-based and differentiable representation that parameterizes both watertight and non-watertight meshes of arbitrary topology. Our new representation, called *Ghost-on-the-Shell* (**G-SHELL**), enables two important applications: differentiable rasterization-based reconstruction from multiview images and generative modelling of non-watertight meshes. We empirically demonstrate that **G-SHELL** achieves state-of-the-art performance on non-watertight mesh reconstruction and generation tasks, while also performing effectively for watertight meshes.

## 1   INTRODUCTION

The creation of high-fidelity 3D virtual worlds requires a representation of 3D shape that can be rendered and simulated efficiently and realistically. Most commonly, 3D shapes are represented as meshes for which modern graphics pipelines are highly optimized. Because the manual creation of 3D mesh assets is time-consuming, research has focused on the automatic creation from images or generative models. While much of the recent work has focused on watertight meshes [55, 56, 60, 67],

many 3D objects, such as clothing, paper or leaves, are non-watertight, open[1] and thin. The capture and generative modeling of such surfaces is relatively underexplored.

Existing modeling methods for non-watertight meshes typically build an unsigned distance field (UDF) [35, 38], a scalar field of the absolute distances of 3D coordinates to the nearest surface. With UDF, one may obtain non-watertight meshes by extracting and discretizing the zero UDF levelset. However, isosurface extraction from a UDF, compared to that of signed distance fields (SDF), is a non-trivial task: the bisection search strategy in classical algorithms such as Marching Cubes [39] does not simply apply to UDFs. Common workarounds include 1) post-processing double-layered watertight mesh [48, 70], introducing local pseudo-signs for bisection search [15], and using implicit-free point-to-mesh reconstruction methods [10]. These methods inevitably introduce modeling errors, posing a challenge for non-watertight mesh reconstruction and generation.

We take a different approach to modeling non-watertight meshes with the following key observation – most open surfaces can be viewed as entities floating on watertight surfaces, analogous to continents floating on the Earth's surface. In other words, it suffices to model the open surface boundary on some watertight surface template. To formalize this idea, we define a manifold signed distance field (mSDF) on the watertight template, in which the sign indicates whether a point lies in the open surface or not, and the absolute scale indicates the geodesic distance to the boundary. An open surface can now be extracted via isoline extraction with mSDF.

We follow this intuition and design a general representation, *Ghost-on-the-Shell* (dubbed **G-SHELL**), which jointly parameterizes the watertight template and the non-watertight mesh living on it. Specifically, we discretize the 3D space into a grid of cells, of which the vertices store both SDF and mSDF values, and then apply Marching-Cubes-like extraction to obtain the SDF isosurface and the mSDF isoline. Our implementation exploits an efficient mesh extraction algorithm instead of following the naïve two-stage approach of "isosurface to isoline". In a nutshell, we adapt the look-up table in Marching-Cubes-like algorithms, which enumerate all possible configurations of isosurfaces in each cell according to both SDF and mSDF signs. Such an implementation effectively reduces the number of mesh faces created and thus the computational cost of the grid-to-mesh mapping.

Since **G-SHELL** uses only simple, deterministic and parallelizable operations for mesh extraction, mesh-based inverse rendering can now be applied to non-watertight meshes due to nicely-behaved optimization landscapes (with simple Marching-Cubes-like extraction) and memory-friendly computation (with efficient mesh rasterizers). This efficient mesh rasterization means that we are now able to optimize both material and lighting from pixel information by exploiting physics-based rendering of meshes. Furthermore, the regular grid structure of **G-SHELL** allows the extension of recent generative methods, such as diffusion models, to non-watertight meshes for the first time.

In summary, our major contributions are listed below:

- **Mesh representation**. **G-SHELL** is a differentiable representation that effectively parameterizes both watertight and non-watertight meshes of different shape topologies.
- **Efficiency**. With the designed mesh extraction algorithm for **G-SHELL**, we achieve fast reconstruction of non-watertight meshes with differentiable rasterizers. Specifically, we design an efficient mesh extraction algorithm for **G-SHELL**.
- **Physics-based inverse rendering**. **G-SHELL** enables joint optimization of topology, material, and lighting of both watertight and non-watertight meshes.
- **Mesh generation**. The grid parameterization and efficient mesh extraction of **G-SHELL** enables effective generative modeling of both watertight and non-watertight meshes with diffusion models.
- We qualitatively and quantitatively compare **G-SHELL** with current popular 3D representations on reconstruction from realistic images and unconditional generation of both watertight and non-watertight meshes, demonstrating the superiority of **G-SHELL** in representing general 3D shapes.

## 2 RELATED WORK

**Mesh parameterization and extraction**. In contemporary computer graphics pipelines and software, 3D meshes serve as a crucial and foundational representation. The means of mesh reconstruction can mostly be classified into three categories: from mesh templates, from point clouds and from implicit

---

[1]*i.e.*, bordered. Not to be confused with the mathematical term of openness.

fields. Mesh templates enable the optimization of vertex positions to align with object surfaces but can be inflexible for diverse topologies [16, 64]. Some methods [45, 59, 61] utilize remeshing [5, 20] to accommodate topology changes, but careful initialization remains essential to avoid bad local minima during optimization. Point clouds, compared to mesh templates, can be directly obtained from Lidar scans and at the same time offer flexibility in capturing diverse shape topologies. Such flexibility comes at the cost of the challenge in inferring point connectivity – whether two points are adjacent to each other on the target surface. Common methods for building surfaces from point clouds, such as Ball-Pivoting [3] and Delaunay Triangulation [28], are not only slow due to non-parallelizable operations but susceptible to noises in source point clouds.

It is therefore more common to extract meshes from implicit fields, which may be built from point clouds with Poisson reconstruction [24, 46, 54], or from multiview images [41, 67]. Subsequently, meshes can be extracted by identifying and triangulating the zero levelsets with methods like Marching Cubes [39], Marching Tetrahedra [58] and Dual Contouring [23]. Many of these algorithms are made differentiable, such as in Deep Marching Cubes [30], Neural Dual Contouring [9], MeshSDF [49], DMTet [55] and FlexiCubes [56]. However, most of them only apply to watertight meshes due to the use of SDF. To handle non-watertight meshes, some papers propose differentiable methods using UDF. For instance, MeshUDF [15] computes pseudo-signs on grid vertices – the signs of inner products between UDF gradients on grid vertices – and reuses Marching Cubes to extract meshes from the resulted pseudo SDFs. These methods are sensitive to input noises, due to the challenge in locating zero levelsets from non-negative fields. While implicit representations other than UDF exist, such as variants of generalized winding number [2, 11, 21] and DeepCurrents [44], there is no efficient differentiable mesh parameterization for them. In comparison, our method robustly and efficiently models non-watertight meshes by extracting zero levelsets in a Marching-Cubes-like manner.

**Differentiable inverse rendering**. It is popular in recent years to perform differentiable inverse rendering through implicit representations, such as NeRF [41] and SDF [60, 67], with which one may utilize differentiable volumetric rendering [37, 41] or surface rendering [22, 32, 67] methods. These methods are adapted to reconstruct non-watertight surfaces with UDF (*e.g.*, NeuralUDF [38], NeUDF [35]) and SDF (*e.g.*, NeAT [40]). However, rendering with implicit representations typically requires multiple and likely expensive queries for each pixel in the rendered image. And since the geometry and color are encoded in an implicit way, it is relatively hard to disentangle geometry, material and lighting during inverse rendering. In contrast, explicit representations, such as point clouds and meshes, can be efficiently rendered with rasterization [25, 33], and allow easy disentanglement of physical properties [17, 34, 42]. Compared to the implicit-based methods for non-watertight mesh modeling, our method takes advantage of mesh-based rasterization to enable fast joint optimization of shapes, materials and lighting from multi-view images.

**Generative modeling of geometry**. 3D generation is widely studied for various representations, including explicit representations like mesh [12, 13, 36, 43], point clouds [65, 68], voxels [62], and implicit ones like signed distance functions (SDFs) [6, 69] and neural radiance fields (NeRF) [6, 7, 26, 53]. However, apart from few mesh-based methods, all the other ones do not directly generate meshes with arbitrary topology. As a result, one typically has to perform additional post-processing steps to extract meshes, which can be time-consuming and may introduce additional errors.

More recently, methods have been proposed to generate meshes using intermediate grid representations either through direct 3D modeling [12, 13, 36] or through lifting information from 2D generative models [8, 31]. While these methods achieve success in generating watertight 3D meshes, none of them can generate non-watertight meshes. It is also possible to directly generate meshes in an autoregressive way: for instance, PolyGen [43] builds a transformer-based autoregressive model to alternately produce vertices and edges. However, autoregressive models can be so flexible that they hardly scale to complex meshes (especially those not created by designers) with a very dense set of vertices and often produce self-intersecting meshes. Our method, instead, is capable of generating both watertight and non-watertight meshes with fine geometric details and without self-intersections.

## 3 PRELIMINARIES: SDF-BASED MESH EXTRACTION

We briefly summarize Marching Cubes, a classical SDF-based mesh extraction method, and introduce some of its variants. In a nutshell, Marching Cubes discretizes the 3D space with a 3D cubic grid and extracts faces from each cubic cell with a simple linear assumption: the SDF value of any point in each cell is a barycentric interpolation of those on cell corners. Specifically, given any point $x$ in

a cell with 8 corners $(p_1, p_2, ..., p_8)$, we first obtain the barycentric coordinate $(c_1, c_2, ..., c_8)$ such that $x = \sum_i c_i p_i$. Under the linear assumption, the extracted mesh must be polygons with their vertices on the cubic cell edges. Therefore, it suffices to compute the mesh vertex position $p'$ on each edge $(p_i, p_j)$ with SDF values $s_i < 0 < s_j$, respectively. The vertex position is simply a linear interpolation between $p_i$ and $p_j$: $u = (s_i p_j - s_j p_i)/(s_i - s_j)$.

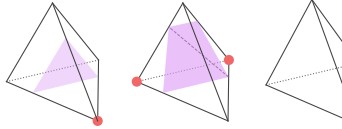

The connectivity of these extracted mesh vertices can be efficiently inferred through a look-up table. One may instead use tetrahedral grids, leading to the variant called Marching Tetrahedra, for which we visualize the look-up table in Figure 2. Since the vertices are computed through simple differentiable operations, one may parameterize watertight meshes with a potentially deformable grid of SDF values – *e.g.*, deep marching cubes (with some relaxation) [30] and DMTet [55].

Figure 2: Look-up table for Marching Tetrahedra (up to rotation symmetry). Grid vertices with and without a red dot possess SDF values of opposite signs.

## 4 G-SHELL: AN EXPRESSIVE REPRESENTATION OF GENERAL 3D SHAPES

### 4.1 OPEN SURFACE LIVING ON WATERTIGHT SURFACE

We start with the following simple observation on a category of open surfaces, which guides our insight to parameterize general 3D shapes with open surfaces that live on a watertight surface.

> Any smooth and simply-connected open surface can be smoothly deformed to be a subset of a sphere.

This is a direct consequence of classical topological theories on surfaces, of which more mathematical details are given in Appendix A. Indeed, a large number of surfaces (*e.g.*, plain T-shirts) can be completed to a watertight surface by first contracting the holes and later deforming it into a sphere. Inspired by this observation, we define a continuous and differentiable mapping $\nu : \mathcal{M} \to \mathbb{R}$ on the template sphere $\mathcal{M}$ to characterize if a point belongs to the open surface $\mathcal{M}_o$:

$$\underbrace{\nu(x) > 0, \ \forall x \in \text{Interior}(\mathcal{M}_o),}_{\text{Case 1: inside the open surface}} \quad \underbrace{\nu(x) = 0, \ \forall x \in \partial\mathcal{M}_o,}_{\text{Case 2: on the surface boundary}} \quad \underbrace{\nu(x) < 0, \ \text{Otherwise},}_{\text{Case 3: outside the open surface}}$$

where $\nu$ can be instantiated as the signed geodesic distance to the open surface boundary living on the watertight template. While the number of choices of $\nu$ given some $\mathcal{M}$ and $\mathcal{M}_o$ can be infinite, without loss of generality we call the field of $\nu$ manifold signed distance field (mSDF), since it is defined on a manifold surface and characterizes the boundary in a way that is similar to SDF.

The problem now effectively reduces to learning a "2D mesh" defined by the zero isoline of $\nu$. Just as 3D meshes on the zero isosurface can be parameterized by 3D cubic grids in deep marching cubes, "2D meshes" (polygonal curve) can also be parameterized by a "2D grid", *i.e.*, a mesh for the (deformed) sphere: simply to learn a $\nu$ value on each of the sphere mesh vertices from which we extract non-watertight meshes. This process is illustrated in Figure 3. Intuitively speaking, an open surface can be viewed as the remaining essence after cutting out the hollow vacuum on a 3D shell, and hence we name the proposed 3D representation Ghost-on-the-Shell[2].

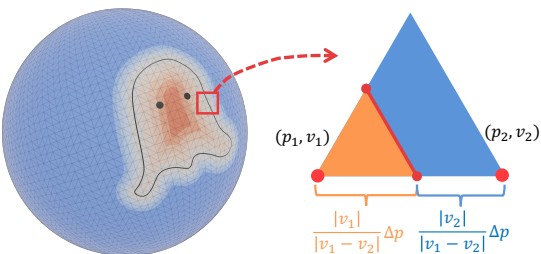

Figure 3: Illustration of non-watertight mesh extraction from some watertight triangular mesh. $p_1, p_2$ are the positions of (watertight) mesh vertices. $\Delta p = \|p_1 - p_2\|$ and $\nu_1 > 0 > \nu_2$ are the corresponding mSDF values. The orange triangle is extracted and the blue polygon is discarded.

Such a naïve construction, however, poses modeling challenges when applied to general objects. First, it cannot capture watertight surfaces that are not homeomorphic to spheres (*e.g.*, donuts). Furthermore, some naïve deformation of a surface in 3D may result in self-intersection and therefore addressing this requires additional regularization and/or modeling techniques.

---

[2]Inspiration drawn from the manga series *Ghost in the Shell*

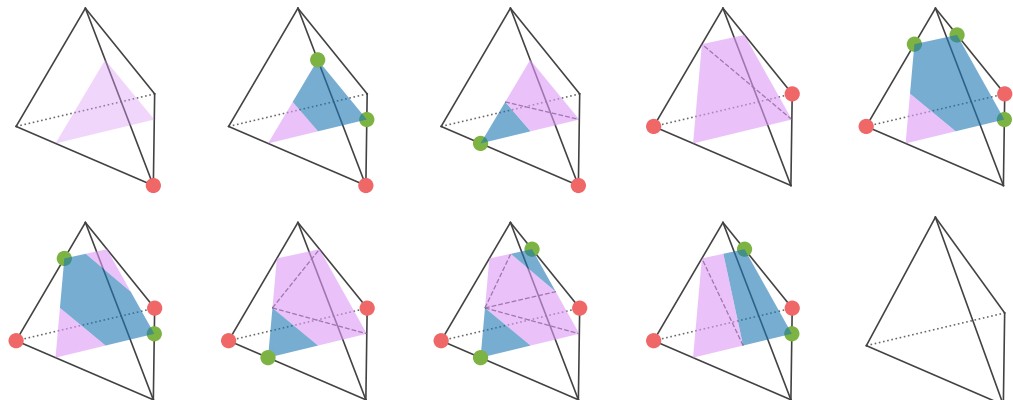

Figure 4: **G-SHELL** look-up table (up to rotational symmetry) for tetrahedral grids. Grid vertices with and without a red dot possess SDF values of opposite signs, and green dots on watertight mesh vertices indicates negative mSDF values. The pink regions represent the final extracted faces while the blue ones are the discarded regions on the watertight template mesh. Colored polygons other than triangles are cut along dashed lines.

Instead, we propose to jointly learn a general watertight mesh template, parameterized by a 3D grid of SDF values, in order to capture a larger set of meshes[3]. As we are not able to define $\nu$ with mesh topology changing over time, we instead define $\nu$ in the 3D space. Specifically, we store the discretized values of $\nu$ in a 3D grid. The mSDF value of any point in a grid cell can therefore be computed by a barycentric interpolation of the $\nu$ values on the grid cell vertices. We note that **G-SHELL** reduces to a typical watertight surface representation if all mSDF values on the grid are set to positive values (*i.e.*, no valid topological hole is defined on the manifold).

### 4.2 EFFICIENT MESH EXTRACTION WITH **G-SHELL**

With SDF and mSDF values stored in the same 3D grid, we obtain for **G-SHELL** an efficient Marching-Cubes-like algorithm which reuses the interpolation coefficient (*rf.* Eqn. 3) for the mSDF sign computation. Specifically, with an edge $(p_i, p_j)$, the corresponding SDF values $s_i < 0 < s_j$ and mSDF values $\nu_i, \nu_j$, we can compute the mSDF value on the extracted mesh vertex as $\nu' = (s_i\nu_j - s_j\nu_i)/(s_i - s_j)$. We give an example of the look-up table for tetrahedral grids in Figure 4, which enumerates all possible cases of SDF signs (on grid vertices) and mSDF signs (on watertight mesh vertices). Despite using tetrahedral grids as an example, we note that **G-SHELL** is generally applicable to other grid structures and not limited to tetrahedral grids.

## 5 APPLICATIONS OF **G-SHELL**

### 5.1 MESH RECONSTRUCTION FROM MULTIVIEW IMAGES

With **G-SHELL**, existing differentiable rasterization-based rendering methods (*e.g.*, [17, 42]) can be seamlessly applied for end-to-end reconstruction of both 3D watertight and non-watertight meshes from multiview RGB and binary mask images. Reconstruction with rasterization not only allows the final geometry to be explicitly optimized without pose-processing, but also saves memory and time compared to volumetric rendering with UDFs [35, 38] - there is no need to evaluate densities on a number of sample points per ray anymore. Moreover, with physics-based mesh rendering, one can jointly optimize geometry, material and lighting in a single stage.

We note, however, some particular difficulties in non-watertight mesh reconstruction from images. Unlike watertight meshes where we never get to see the inside surface, non-watertight surfaces have two sides and one side may be more visible than the other. For example, the inside of a long dress may not be fully observed and, when actually seen, it is also likely to be poorly illuminated. Similarly, indirect illumination has to be considered along with realistic materials (especially highly specular ones) and potentially complex geometry. To simplify the problem while still being able to demonstrate the effectiveness of the **G-SHELL** representation, we use Nvdiffrecmc [17], an occlusion-aware differentiable renderer that ignores indirect illumination but considers shadow rays.

---

[3]Indeed, any orientable open surface without self-intersection can be modeled thereby.

Another technical challenge is how to identify the existence and location of topological holes with only 2D images, particularly when only a limited number of views are available. We therefore propose to regularize the mSDF values of the reconstructed mesh by introducing a "hole-opening" loss (the mSDF is parameterized by a function with some parameter set $\theta_{\text{mSDF}}$):

$$L_{\text{mSDF-reg}}(\theta_{\text{mSDF}}) = \underbrace{\sum_{u:\nu_{\theta_{\text{mSDF}}}(u)\geq 0} L_{\text{huber}}(\nu_{\theta_{\text{mSDF}}}(u))}_{\text{Encourage hole opening}} + \tau \cdot \underbrace{\sum_{\substack{u':\nu_{\theta_{\text{mSDF}}}(u')=0 \\ u' \text{ visible from some } q \in Q}} L_{\text{huber}}(\nu_{\theta_{\text{mSDF}}}(u') - \epsilon)}_{\text{Regularize holes from being too large}} \quad (1)$$

in which $Q$ is the set of training camera poses, $\tau$ and $\epsilon$ are some positive scalars, $L_{\text{huber}}$ is the Huber loss function. We introduce the second regularization term to discourage topological holes from being too large, especially during the early stage of the optimization process. We provide all the details regarding the remaining regularization losses and other training settings in Appendix B.

## 5.2 G-MESHDIFFUSION: GENERATIVE MODELING OF GEOMETRY

With the regular grid structure of the **G-SHELL** parameterization, it is straightforward to train generative models to produce the grid attributes (SDF, mSDF and potentially grid deformation) to enable non-watertight mesh generation. Indeed, **G-SHELL** enables generative modeling with diffusion models [19] in which a regular input structure is necessary.

To demonstrate the generative modeling of **G-SHELL**, we consider MeshDiffusion [36], which generates watertight meshes by sampling SDF and grid deformation in a 3D tetrahedral grid, to generate non-watertight meshes. Although it is possible to simply introduce the additional dimension of mSDF on the grid vertex attributes, the generated shapes can be noisy as pointed out in [36]. Specifically, the boundary vertices of a generated non-watertight mesh are computed via an interpolation with mSDF:

$$u' = \frac{|\nu_1|}{|\nu_1 - \nu_2|} \cdot u_2 - \frac{|\nu_2|}{|\nu_1 - \nu_2|} \cdot u_1, \quad \nu_1 < 0 < \nu_2, \quad (2)$$

in which $(u_1, u_2)$ is an edge on the extracted watertight template and $\nu_1, \nu_2$ are the corresponding mSDF values. Because $\nu_i$ can be any real number, a naïve diffusion loss results in unevenly weighted prediction on $u'$. One could normalize $\nu$ to $\pm 1$, similar to what MeshDiffusion does for the SDF values, but it comes at the cost of expressiveness as the grid deformation is already used to compensate the error resulting from the normalization of SDF values.

We therefore propose a modified version of the MeshDiffusion architecture to generate continuous mSDF values. Specifically, by setting $\alpha = |\nu_1|/|\nu_1 - \nu_2|$, we rewrite Eqn. 2 into a linear mapping of $(u_1, u_2)$: $u' = \alpha u_2 - (1 - \alpha)u_1$. As a result, we may alternatively generate the linear interpolation coefficient $\alpha$, which is bounded in $[0, 1]$. In the case of tetrahedral grids as in DMTet, there are 12 candidate edges $(u_i, u_j)$ in each single cell since $u_i$'s always lie on tetrahedral grid edges. We simply set the diffusion model to generate all $\alpha$'s for these candidates edges in each single cell. The $\alpha$'s to be used are eventually chosen based on the configuration of each generated tetrahedral cell (as in Figure 4). Similar to MeshDiffusion, we collect a dataset of non-watertight meshes by running inverse rendering on multiview datasets of 3D objects. We term our final diffusion model on the **G-SHELL** representation as *G-MeshDiffusion*. More details can be found in Appendix C.

## 6 EXPERIMENTS AND RESULTS

### 6.1 RECONSTRUCTION FROM MULTIVIEW IMAGES

**Baselines**. We compare our method to current state-of-the-art methods for non-watertight mesh reconstruction: NeuralUDF [38], NeUDF [35] and NeAT [40]. We also evaluate watertight reconstruction methods including NeuS [60] and Nvdiffrecmc [17] (with DMTet [55]). We follow the original settings of these baselines, but train them for 400,000 iterations with mask loss weighted by 0.1. During testing, the novel views are rendered and subsequently evaluated at a resolution of 512. For UDF-based methods, the non-watertight explicit meshes are extracted using MeshUDF [15], while for SDF-based methods (NeuS/NeAT), the meshes are extracted with Marching Cubes [39] with the same resolution. For Nvdiffrecmc with DMTet, we use a grid resolution of 128.

**Dataset**. We use DeepFashion3D-v2 [18] to quantitatively evaluate the performance of reconstruction with **G-SHELL** on non-watertight meshes. Specifically, we use ground truth meshes of 9 instances

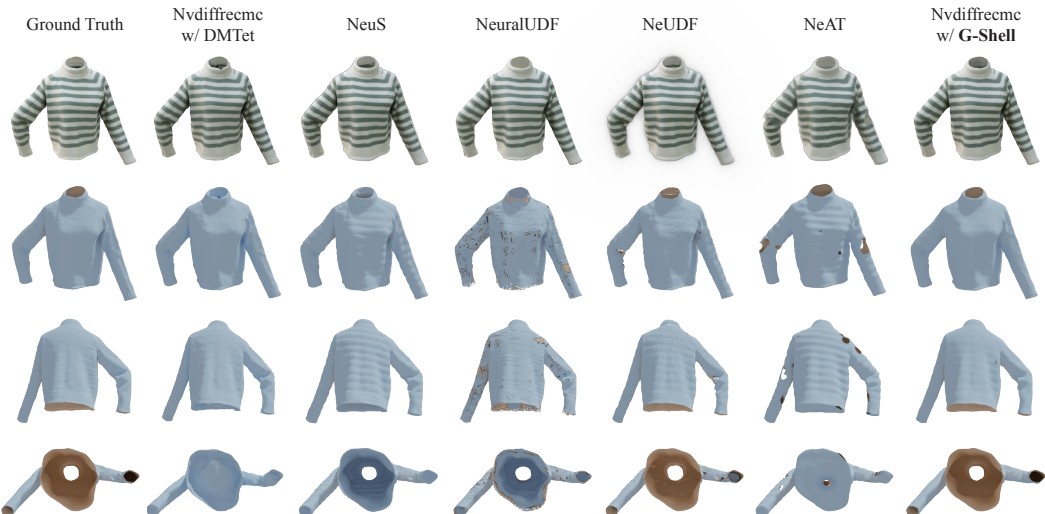

Figure 5: Comparison between reconstruction w/ G-SHELL and other baseline methods on multiview reconstruction on DeepFashion3D dataset. Top row: reconstructed texture. Bottom 3 rows: reconstructed meshes.

| Method \ Instance ID | 30 | 92 | 117 | 133 | 164 | 320 | 448 | 522 | 591 | Avg |
|---|---|---|---|---|---|---|---|---|---|---|
| NeuS [60] | 31.876 | 26.850 | 29.234 | 29.692 | 30.405 | **33.827** | 31.591 | **32.846** | 26.282 | 30.289 |
| Nvdiffrecmc w/ DMTet [55] | 32.570 | 33.542 | 28.143 | 30.814 | 28.781 | 32.336 | 33.917 | 32.144 | 32.872 | 31.677 |
| NeuralUDF [38] | 30.264 | 25.887 | 28.908 | 31.115 | 28.463 | 30.739 | 32.185 | 29.965 | 32.574 | 30.011 |
| NeUDF [35] | 30.312 | 31.957 | 27.448 | 30.275 | 28.324 | 32.568 | 32.511 | 31.371 | 34.898 | 31.074 |
| NeAT [40] | 27.407 | 28.228 | 24.129 | 26.944 | 24.887 | 29.630 | 30.846 | 27.149 | 29.841 | 27.674 |
| Nvdiffrecmc w/ G-SHELL | 33.165 | 33.959 | **30.204** | 33.164 | 31.139 | 33.429 | **34.997** | 32.724 | 34.579 | 33.039 |
| Nvdiffrecmc w/ G-SHELL (FC) | **33.169** | **34.615** | 30.223 | **33.368** | **31.735** | 33.611 | 34.897 | 32.499 | **35.427** | **33.277** |

Table 1: **PSNR** ($\uparrow$) on DeepFashion3D garment instances. Marker: **1st rank** and 2nd rank.

from DeepFashion3D-v2 that overlap with the instances of DeepFashion3D-v1 used in [35, 38]. Different from previous work, such as NeuralUDF, that uses albedo images for training and testing, we instead use images rendered with realistic lighting and shading. Specifically, for each instance we use Blender with Cycles engine and realistic environment lightmap to render 72 views (RGB images and binary segmentation masks) for training and 200 views for testing.

**Settings**. For the multi-view reconstruction with G-SHELL, we set the grid resolution to 128 for tetrahedral grids and 80 for FlexiCubes, and train the models with 5000 iterations using a batch size of 2 views. For all baseline methods, we use the default settings as in their papers. Our experiments are carried out with tetrahedral grid implementation of G-SHELL by default, if not otherwise specified. To demonstrate that the idea of G-SHELL can be quite general, we also evaluate a simple variant of G-SHELL implemented with FlexiCubes [56], denoted by G-SHELL (FC) in Table 1 and Table 2.

**Reconstruction quality**. Table 1 shows the PSNR averaged over all test views. For all shapes fitted with implicit-based baseline methods (*i.e.*, except for Nvdiffrecmc with DMTet), we render images directly from the learned implicit fields instead of the extracted meshes. We also evaluate all methods on the quality of the reconstructed geometry by computing the bi-directional pointcloud-to-mesh Chamfer distance. The results are given in Table 2, from which we see that G-SHELL achieves excellent reconstruction quality and outperforms a number of state-of-the-art methods. Figure 5 gives

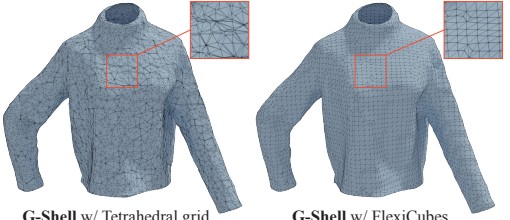

**G-Shell** w/ Tetrahedral grid    **G-Shell** w/ FlexiCubes

Figure 6: Reconstructed mesh topology of G-SHELL with tetrahedral grids and G-SHELL with Flexicubes. Mesh vertex count: Left = 10,658, Right = 6,183.

a qualitative comparison, where the front and back mesh faces (re-oriented to be consistent to their neighbors) are painted in different colors. The qualitative results in Figure 6 demonstrates that the mesh topology reconstructed by G-SHELL with FlexiCubes is more regular than G-SHELL with tetrahedral grids, making it better suited for physical simulation. More results are in Appendix F.

| Method \ Instance ID | 30 | 92 | 117 | 133 | 164 | 320 | 448 | 522 | 591 | Avg |
|---|---|---|---|---|---|---|---|---|---|---|
| NeuS [60] | 0.450 | 1.244 | 0.505 | 0.709 | 0.528 | 0.426 | 0.734 | 0.598 | 1.737 | 0.770 |
| Nvdiffrecmc w/ DMTet [55] | 0.629 | 0.466 | 0.724 | 0.856 | 0.722 | 0.444 | 0.444 | 0.649 | 1.026 | 0.662 |
| NeuralUDF [38] | 0.457 | 0.867 | 0.281 | 0.215 | 0.198 | 0.554 | 0.197 | 0.561 | 0.206 | 0.393 |
| NeUDF [35] | 0.315 | 0.458 | 0.204 | **0.107** | 0.184 | 0.432 | **0.159** | 0.585 | **0.128** | 0.286 |
| NeAT [40] | 0.605 | 0.325 | 1.010 | 0.873 | 0.538 | 0.305 | 0.323 | 0.591 | 0.237 | 0.534 |
| Nvdiffrecmc w/ **G-SHELL** | **0.212** | **0.207** | **0.133** | 0.144 | 0.160 | **0.173** | 0.173 | **0.208** | 0.178 | **0.177** |
| Nvdiffrecmc w/ **G-SHELL** (FC) | 0.235 | 0.227 | 0.146 | 0.154 | **0.165** | 0.229 | 0.195 | 0.261 | 0.234 | 0.203 |

Table 2: **Chamfer distance (cm ↓)** on DeepFashion3D garment instances. Marker: **1st rank** and 2nd rank.

| Method \ Metalness | 0 | 0.2 | 0.4 | 0.6 | 0.8 | 1 | Avg | Method \ Metalness | 0 | 0.2 | 0.4 | 0.6 | 0.8 | 1 | Avg |
|---|---|---|---|---|---|---|---|---|---|---|---|---|---|---|---|
| NeuS [60] | 33.59 | 32.03 | 31.62 | 31.21 | 30.58 | 30.45 | 31.58 | NeuS [60] | 0.47 | 0.59 | 0.50 | 0.57 | 0.68 | 0.72 | 0.59 |
| NeuralUDF [38] | 31.42 | 29.54 | 29.45 | 29.39 | 29.25 | 28.98 | 29.67 | NeuralUDF [38] | 0.51 | 1.06 | 1.05 | 0.98 | 0.53 | 0.85 | 0.83 |
| NeUDF [35] | 32.45 | 29.32 | 29.04 | 29.19 | 29.13 | 29.18 | 29.72 | NeUDF [35] | 0.26 | 0.52 | 0.40 | 0.45 | 0.70 | 0.44 | 0.46 |
| NeAT [40] | 28.41 | 27.37 | 28.07 | 26.93 | 26.58 | 26.85 | 27.37 | NeAT [40] | 0.59 | 0.65 | 0.61 | 0.68 | 0.70 | 0.73 | 0.66 |
| **G-SHELL** | **36.01** | **34.23** | **32.86** | **33.08** | **32.93** | **32.31** | **33.57** | **G-SHELL** | **0.20** | **0.22** | **0.28** | **0.24** | **0.24** | **0.27** | **0.24** |

Table 3: Ablation study on metallic materials. We use Nvdiffrecmc along with the proposed **G-SHELL** for our reconstruction in the experiment. Left: **PSNR (↑)**. Right: **Chamfer distance (cm ↓)**.

From Figure 5, we can observe that our method achieves much better prediction on novel views than the other baselines due to the better modeling of lighting, occlusion and material. Due to the highly concave structure in the clothing interior, we can also see that watertight reconstruction baselines often fail to reconstruct the inner side of clothing in the DeepFashion3D dataset.

**Efficiency in training and inference**. In addition, we compare the runtime of all non-watertight methods (tested on the same machine with a single NVIDIA RTX 6000 GPU). Along with Nvdiffrecmc, **G-SHELL** takes only **3 hours** to fit a ground truth shape while NeuralUDF, NeUDF and NeAT take 17.3, 16.4, and 4.3 hours, respectively. For novel-view synthesis with all images rendered with a resolution of $512 \times 512$, our method runs at **2.7 sec/img** (inferring from a learned tetrahedral grid with the Nvdiffrast rasterizer [27]), while NeuralUDF, NeUDF and NeAT run at 1.8 min/img, 1.4 min/img, and 9.7 min/img, respectively. Compared to the other methods, ours is significantly faster in both training and inference due to its highly efficient rasterization.

## 6.2 HYBRID WATERTIGHT AND NON-WATERTIGHT MESH RECONSTRUCTION

To demonstrate that **G-SHELL** may be used for reconstructing both watertight and non-watertight shapes at the same time, we test our method on an instance from the synthetic NeRF dataset [41] in which all the images are taken from the objects above and never below. We adopt a minimalist assumption that unseen regions should always be empty and reconstruct the target shapes with **G-SHELL**. On the left of Figure 7, we show that the upper surface of the target shape is well reconstructed without adding any unnecessary lower surface faces. On the right hand side of Figure 7, we also visualize the generalized winding number field [2, 21] by randomly sampling points near the surface to quantify local manifoldness. It shows that in-

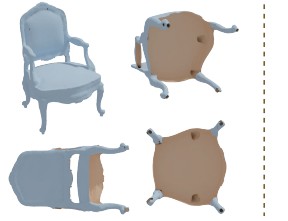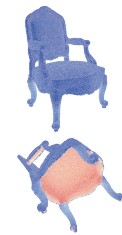

Figure 7: Left: the reconstructed mesh, with back faces painted orange. Right: generalized winding number field; points near the mesh boundary are colored white.

verse rendering with **G-SHELL** can reconstruct a hybrid shape of both watertight and non-watertight parts where the regions with only visible upper surface are reconstructed as single-layered meshes.

## 6.3 RECONSTRUCTION UNDER SPECULAR LIGHTING

One key advantage of mesh inverse rendering is that geometries may be reconstructed in cases of complex lighting and materials. To demonstrate such an advantage, we conduct ablation experiments and compare the performance of non-watertight mesh reconstruction methods on shapes with specular surfaces. Specifically, we modify the material of the sweater mesh presented in Figure 5 by setting the roughness parameter to $0.4$ and create a set of meshes with the metalness parameter ranging from 0 to 1. We use the same set of hyperparameters for the experiment. Results in Table 1 and Table 3 verify that **G-SHELL** can well reconstruct both images and shapes under complex lighting and materials.

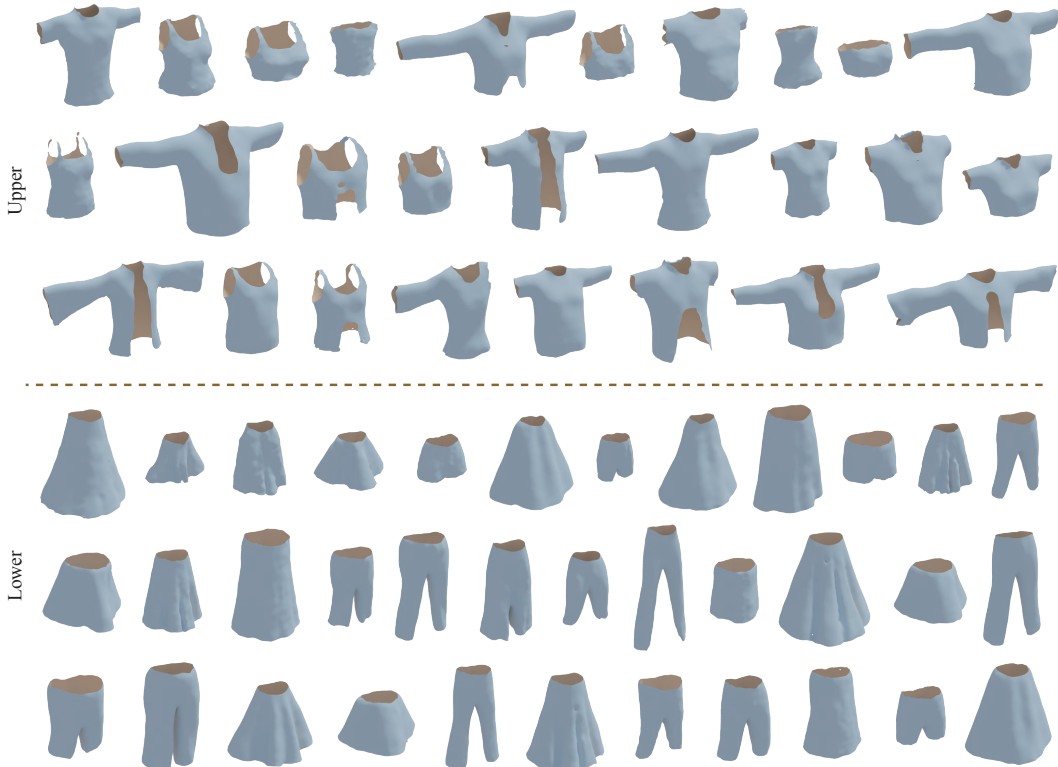

Figure 8: Samples of unconditionally generated upper and lower outfits by G-MeshDiffusion.

| | Method | MMD ($10^{-3}$, ↓) | | COV (%, ↑) | | 1-NNA (%, ↓) | | MV-FID (↓) |
|---|---|---|---|---|---|---|---|---|
| | | CD | EMD | CD | EMD | CD | EMD | |
| Lower Garments | MeshDiffusion [36] | 1.88 | 68.72 | 38.01 | 31.79 | 88.99 | 91.03 | 191.09 |
| | GET3D [12] | 1.57 | 57.19 | **46.00** | **48.13** | 79.66 | 72.82 | 95.69 |
| | G-MeshDiffusion | **1.36** | **55.30** | 41.92 | 41.03 | **68.29** | **67.23** | **79.01** |
| Upper Garments | MeshDiffusion [36] | 1.24 | 55.37 | **51.51** | 46.71 | 77.44 | 78.77 | 167.69 |
| | GET3D [12] | 1.37 | 56.86 | 48.85 | 43.87 | 84.37 | 80.11 | 112.45 |
| | G-MeshDiffusion | **1.05** | **51.15** | 51.33 | **49.20** | **61.37** | **60.66** | **88.98** |

Table 4: Quantitative results of the proposed **G-SHELL** diffusion model (G-MeshDiffusion).

## 6.4 GENERATIVE 3D MESH MODELING

**Baselines**. As there are no mesh generative models for open surfaces with manifold structures, we compare our method with two watertight mesh generative models: MeshDiffusion [36] and GET3D [12], both using the SDF-based DMTet [55] as the representation. Detailed training settings are deferred to Appendix C. We implement **G-SHELL** with tetrahedral grids in this experiment.

**Dataset**. We collect meshes in 4 categories (T-shirt, Top, Skirts, Trousers) from the Cloth3D dataset [4] and regroup them into two new categories: upper garments (including T-shirt and Top) and lower garments (including Skirts and Trousers). For both MeshDiffusion (using the DMTet representation) G-MeshDiffusion (using the **G-SHELL** representation), we run inverse rendering on meshes with known environment lightmaps and known materials using RGB, binary mask, and depth supervision. We generally follow the same settings of [36] for G-MeshDiffusion. For GET3D, we follow the same training setting as [12] and render multiview RGB images for training.

**Evaluation metrics**. For each model, we sample a set of meshes, with the size of the test sets, using 100 steps of DDIM [57] and apply standard Laplacian smoothing to these meshes. Similar to [12, 36], we evaluate point cloud metrics between the point clouds sampled from generated meshes and those from ground truth meshes. To compensate the lack of perceptual measure in the point cloud metrics, we also evaluate the generated results with multiview FID (MV-FID) [36, 69], which is computed by an average of FID (Fréchet Inception Distance) scores of 20 views (rendered with fixed light sources

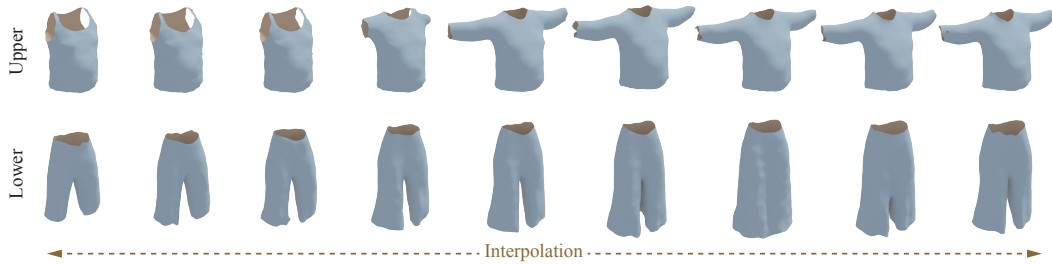

Figure 9: Interpolation results of our generated samples.

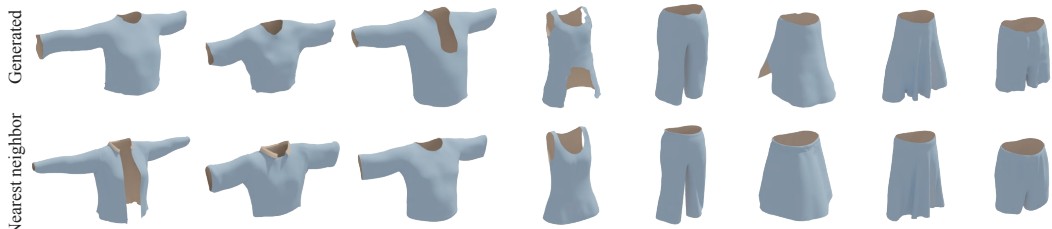

Figure 10: Nearest neighbor of our generated sample in the training set.

and a diffuse-only mesh material). During rendering, we do not re-orient the face normals towards the camera so that the difference between watertight and open surfaces can be taken into account.

**Qualitative and quantitative results**. The quantitative results are given in Table 4. We observe that G-MeshDiffusion generally achieves better performance than the watertight mesh generation methods (MeshDiffusion and GET3D), but more importantly, G-MeshDiffusion can better capture the single-sided nature of non-watertight meshes as it achieves a significantly better MV-FID score. In addition, we qualitatively show some unconditionally sampled meshes from G-MeshDiffusion in Figure 8. Following [36], we also provide some interpolation sequences (obtained by spherical linear interpolation with the initial Gaussian noises using 100-step DDIM inference) in Figure 9. The interpolation results demonstrate a smooth transition across different clothing styles. Finally, we show in Figure 10 the nearest neighbor of our generated meshes in the training set. The results show that G-MeshDiffusion does not memorize the training samples and can generate novel shapes.

## 7 DISCUSSION ON LIMITATIONS

**Representation**. **G-SHELL** is not able to model shapes with self-intersections. Nor does it model non-orientable surfaces such as Möbius strips, since the use of SDF implies orientability. Besides, compared to implicit-based methods of which the resolution can be viewed as "infinite", **G-SHELL** generally may require a higher resolution in order to model tiny topological holes.

**Mesh reconstruction**. The discontinuity in mesh rendering poses a severe optimization problem when the target geometry is complex, since the entanglement between geometry and indirect illumination comes into play. In this paper, we only take shadow rays into consideration, but in more complex scenarios one might need to model indirect illumination.

**Mesh generation**. 3D U-Nets are not memory-efficient, and it is hard to scale them to high resolutions. Future work for better generative modeling with **G-SHELL** may include more efficient architectures such as those using triplane features [6], and alternative 3D generative methods similar to GET3D [12].

## 8 CONCLUDING REMARKS

Our proposed **G-SHELL** is a general 3D mesh representation which models both watertight and non-watertight meshes. By introducing the new quantity of manifold signed distance field (mSDF) on a learnable watertight mesh template with varying topology, we enable both rasterization-based reconstruction and diffusion-model-based unconditional generation of non-watertight meshes. Such a design leads to better performance and greater flexibility in non-watertight mesh modeling. Still, **G-SHELL** is only able to parameterize a limited subset of meshes, especially when limited computational resources are available. Furthermore, it calls for better inverse rendering and generative modeling techniques to fully leverage the flexibility of **G-SHELL**.

## ACKNOWLEDGEMENT

We sincerely thank Peter Kulits for paper proofreading and Zhouyingcheng Liao for creating physical simulation demos. Authors listed as equal contribution (*resp.*, shared last author) are allowed to switch their orders in the author list in their resumes and websites. The paper title was proposed during an after-dinner coffee chat among Zhen Liu, Yao Feng, Yuliang Xiu, Weiyang Liu and Tim Xiao, especially due to Yuliang Xiu (for proposing "Shell") and Weiyang Liu (for the connection to the manga series *Ghost in the Shell* [4]).

**Disclosure**. This work was supported by the German Federal Ministry of Education and Research (BMBF): Tübingen AI Center, FKZ: 01IS18039B, and by the Machine Learning Cluster of Excellence, EXC number 2064/1 – Project number 390727645. MJB has received research gift funds from Adobe, Intel, Nvidia, Meta/Facebook, and Amazon. MJB has financial interests in Amazon, Datagen Technologies, and Meshcapade GmbH. While MJB is a part-time employee of Meshcapade, his research was performed solely at, and funded solely by, the Max Planck Society. LP is supported by the Canada CIFAR AI Chairs Program and NSERC Discovery Grant. WL was supported by the German Research Foundation (DFG): SFB 1233, Robust Vision: Inference Principles and Neural Mechanisms, TP XX, project number: 276693517. YF is partially supported by the Max Planck ETH Center for Learning Systems. YX is funded by the European Union's Horizon 2020 research and innovation programme under the Marie Skłodowska-Curie grant agreement No.860768 (CLIPE).

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

# Appendix

## Table of Contents

## A  SOME MATHEMATICS ON SURFACES

We first state the condition for homeomorphism between compact triangulable bordered surfaces[5]:

**Theorem 1 ([50])** *Two compact triangulable bordered surfaces are homeomorphic if and only if they both have the same number of boundary curves, the same Euler characteristic, and are either both orientable or else both non-orientable.*

It is apparent that any simply-connected surface is homeomorphic to a compact disk on a sphere, since simply-connected bordered surfaces are triangulable.

Indeed, one may establish a more general relation between compact bordered surfaces and closed surfaces with the following classification theorem:

**Theorem 2 (Classification Theorem of Surfaces [29])** *Every compact surface (without boundary) is homeomorphic to one of the following:*

- *The sphere $\mathbb{S}$.*

- *A connected sum of tori: $\mathbb{T}\#...\#\mathbb{T}$.*

- *A connected sum of projective planes: $\mathbb{P}^2\#...\#\mathbb{P}^2$*

Together with Theorem 1, one may conclude that any connected orientable bordered surface (which is triangulable) is homeomorphic to a compact bordered surface on either a sphere or a connected sum of tori.

We conclude this section by noting the following theorem:

**Theorem 3 ([1])** *Every bordered surface $\overline{\mathcal{F}}$ can be regularly embedded in a surface. If $\overline{\mathcal{F}}$ is compact, it can be regularly embedded in a closed surface.*

---

[5]*i.e.*, 2-manifolds, as opposed to the casual daily-life definition of surfaces.

# B    DETAILED SETTINGS ON MESH RECONSTRUCTION

## B.1    LOSSES

**RGB and mask supervision**. Given multiview RGB images $I_{GT}$ and the corresponding binary masks $M_{GT}$ (to segment foreground from background), we optimize the following losses:

$$L_{RGB} = \|I_{RGB, \text{ Rendered}} \odot M_{GT} - I_{RGB, \text{ GT}} \odot M_{GT}\|_2^2, \tag{3}$$

$$L_{mask} = \|M_{\text{Rendered}} - M_{GT}\|_2^2. \tag{4}$$

Since the direct output of the renderer is a high-dynamic range (HDR) image $\hat{I}_{RGB, \text{ Rendered}}$, we follow [17] and use the same tonemapper $\Gamma$ to map the HDR image into a sRGB image with pixels in the range $[0, 255]$:

$$I_{RGB, \text{ Rendered}} = T(\hat{I}_{RGB, \text{ Rendered}}) = \Gamma(\log(\hat{I}_{RGB, \text{ Rendered}} + 1)), \tag{5}$$

where $\Gamma$ is the sRGB transfer function [17].

**mSDF supervision**. Since all pixels with $M_{GT} = 0$ should have a non-positive projected mSDF value we introduce the following loss on mSDF values:

$$L_{mSDF} = \|(M_{\nu} - M_{GT})_+ \odot sgn(1 - M_{GT})\|_1, \tag{6}$$

in which $M_{\nu}$ is the rendered (*i.e.*, projected) mSDF image.

**Depth supervision**. In cases when ground truth depth images are provided, we include a depth supervision loss:

$$L_{\text{Depth}} = \|I_{\text{Depth, rendered}} - I_{\text{Depth, GT}}\|_1. \tag{7}$$

**Eikonal regularization**. To reconstruct smoother surfaces, we use the standard Eikonal regularization [14] for MLP-parameterized SDF values on the vertices of the reconstructed mesh:

$$L_{\text{Eikonal}} = \underset{v \in V}{\mathbb{E}} (\|\nabla f_\theta(v)\|_2 - 1)^2 \tag{8}$$

in which $V$ is the set of vertices of the extracted mesh.

**mSDF regularization**. We rewrite the mSDF regularization, introduced in the main paper, into the following (with $\epsilon = 0.001$):

$$L_{\text{mSDF-reg-open}}(\theta_{\text{mSDF}}) = \sum_{u:\nu_\theta(u) \geq 0} L_{\text{huber}}(\nu_\theta(u)), \tag{9}$$

$$L_{\text{mSDF-reg-close}}(\theta_{\text{mSDF}}) = \sum_{\substack{u':\nu_\theta(u')=0 \\ u' \text{ visible from some } q \in Q}} L_{\text{huber}}(\nu_\theta(u') - \epsilon). \tag{10}$$

To determine if a mesh vertex $u'$ is visible for the set of all training camera poses is computationally costly. Therefore, we instead use an approximated loss for $L_{\text{mSDF-reg-close}}$:

$$L_{\text{mSDF-reg-close}}(\theta_{\text{mSDF}}) = \sum_{\substack{u':\nu_\theta(u')=0 \\ u' \text{ visible from some } q \in Q_t}} L_{\text{huber}}(\nu_\theta(u') - \epsilon). \tag{11}$$

in which $Q_t$ is the batch of camera poses at training iteration $t$.

Note that we stop the gradient flow from these two losses to both SDF and grid node offsets. While it is possible to allow gradients from mSDF regularization to update SDF and grid node offsets as well, we choose not to do so to stabilize the optimization process.

We follow [17] and use the following regularization losses:

**Albedo smoothness regularization**. To effectively decouple light from material, smoothness regularization is imposed on albedo $k_d$:

$$L_{k_d} = \frac{1}{x_{\text{surf}}} \sum_{x_{\text{surf}}} |k_d(x_{\text{surf}}) - k_d(x_{\text{surf}} + \epsilon_{k_d})|, \tag{12}$$

in which $\epsilon_{k_d} \sim \mathcal{N}\left(0, \sigma_{k_d}^2\right)$ (we use $\sigma_{k_d} = 0.01$).

**SDF regularization**. The following SDF regularization is to reduce floaters and inner geometry:

$$L_{\text{SDF-reg}} = \sum_{i,j \in S} H(\sigma(s_i), sgn(s_j)) + H(\sigma(s_j), sgn(s_i)) \tag{13}$$

where $S$ is the set of unique edges on the 3D grid, $H$ is the binary cross-entropy loss, $s_i, s_j$ are the SDF values of the corresponding grid vertices and $sgn$ is the sign function. This regularization loss aims to smooth the SDF in the whole space; as a result, isolated tiny meshes are less likely. This regularization term is complimentary to the Eikonal loss, as Eikonal loss is mostly concerned with the surface smoothness as it is computed on the surface only.

**Monochrome regularization**. The following loss[6] helps decoupling lighting, geometry and material:

$$L_{\text{light}} = \|T(Y(c_d + c_s))) - T(V(I_{\text{ref}}))\|_1 \tag{14}$$

where $c_d$ and $c_s$ are the diffuse and specular light components, $I_{\text{ref}}$ is the target reference image, $T$ is the tonemapping function introduced earlier, $Y(x)$ is the simple luminance operator by averaging RGB channels $(x_r + x_g + x_b)/3$, and $V = \max(x_r, x_g, x_b)$. With this loss term, the demodulated lighting is assumed to be mostly monochrome, *i.e.*, $Y(x) \approx V(x)$.

In addition, we penalize the specular component with the following loss:

$$L_{\text{specular}} = \frac{\lambda_{\text{specular}} \|Y(c_s)\|_1}{\min\{\|Y(c_d)\|_1, \epsilon\}}. \tag{15}$$

**Mesh topology regularization**. Finally, for the **G-SHELL** variant with FlexiCubes, we follow [56] and use the proposed mesh topology regularization $L_{\text{dev}}$ which penalizes mean absolute deviation between the primal and dual vertices in FlexiCubes. The precise loss and parameter definitions are beyond the scope of the paper and we do not elaborate them here.

**Total loss**. In summary, the total loss is

$$L_{\text{Total}} = L_{\text{rendering}} + L_{\text{geometry-reg}} + L_{\text{material-light-reg}} \tag{16}$$

with

$$\begin{aligned} L_{\text{rendering}} = L_{\text{RGB}} &+ \gamma_{\text{Mask}} L_{\text{Mask}} + \gamma_{\text{Depth}} L_{\text{Depth}} \\ &+ \gamma_{\text{mSDF}} L_{\text{mSDF}}, \end{aligned} \tag{17}$$

$$\begin{aligned} L_{\text{geometry-reg}} = \gamma_{\text{Eikonal}} L_{\text{Eikonal}} &+ \gamma_{\text{SDF-reg}} L_{\text{SDF-reg}} \\ &+ \gamma_{\text{mSDF-reg-open}} L_{\text{mSDF-reg-open}} + \gamma_{\text{mSDF-reg-close}} L_{\text{mSDF-reg-close}} \\ &+ \gamma_{\text{dev}} L_{\text{dev}}, \end{aligned} \tag{18}$$

and

$$L_{\text{material-light-reg}} = \gamma_{\text{light}} L_{\text{light}} + \gamma_{\text{specular}} L_{\text{specular}} + \gamma_{k_d} L_{k_d}. \tag{19}$$

**Choice of loss scales**. For DeepFashion3D instances, we set $\gamma_{\text{Mask}} = 1, \gamma_{\text{mSDF}} = 0.5, \gamma_{\text{mSDF-reg-close}} = 10^{-6}/\rho, \gamma_{\text{mSDF-reg-open}} = 3 \times 10^{-6}/\rho, \gamma_{\text{light}} = 0.15$ and $\gamma_{\text{specular}} = 0.0025$, where $\rho = (\text{grid-resolution}/64)^3$. We use a linear schedule for Eikonal regularization: $\gamma_{\text{Eikonal}} = 0.3$ for the first 500 iterations, $\gamma_{\text{Eikonal}} = 0.1$ for iteration 500 to 2000 and $\gamma_{\text{Eikonal}} = 0.01$ for the remaining iterations.

For the synthetic chair experiment, we set $\gamma_{\text{Eikonal}} = 5e - 3, \gamma_{\text{mSDF-reg-close}} = 2e - 4/\rho$. We set $\gamma_{\text{mSDF-reg-open}}$ to $2e - 5/\rho$ for the first 1500 iterations and $2e - 6/\rho$ for the remaining ones.

We set $\gamma_{\text{dev}} = 0.25$ for the **G-SHELL** variant with FlexiCubes but otherwise 0. All other loss scales not mentioned are set to default values as in [17].

---

[6]There are slight differences between the Nvdiffrecmc paper, the official repository of Nvdiffrecmc and our implementation. Please see Sec. B.2 for more details.

### B.2 DIFFERENCES IN IMPLEMENTATIONS FOR LIGHTING REGULARIZATION

We note the difference between the Nvdiffrecmc paper [17] and the official implementation[7]. In the original paper, the lighting regularization is written such that $T$ is the first to be applied to the rendered and reference image:

$$\|Y(T(c_d + c_s))) - V(T(I_{\text{ref}}))\|_1 . \tag{20}$$

The official implementation uses a modified regularization loss, in which the first term tends to encourage specular components ($\epsilon = 0.001$ in the repo):

$$\lambda_{\text{diffuse}} \frac{\|Y(c_d)\|_1}{\min\{\|Y(c_d + c_s)\|_1, \epsilon\}} \|T(Y(c_d + c_s)) - I_{\text{ref}}\|_1 + \frac{\lambda_{\text{specular}} \|Y(c_s)\|_1}{\min\{\|Y(c_d)\|_1, \epsilon\}}. \tag{21}$$

### B.3 RENDERING, OPTIMIZATION AND OTHER SETTINGS

For **G-SHELL** with tetrahedral grids, we set the number of rays samples of the Nvdiffrecmc renderer (for Monte-Carlo approximation of the rendering equation) $N_{\text{sample}} = 24$ for better performance on the interior of mehses (*e.g.*, clothing in DeepFashion3D dataset). In practice, setting $N_{\text{sample}} = 12$ leads to faster convergence (approx. 2 hours) without too much sacrifice on reconstruction quality (PSNR values lowered by less than 1). For **G-SHELL** with FlexiCubes, we instead use $N_{\text{sample}} = 16$.

As occlusion heavily depends on shape topology, it will be hard to optimize both lighting and shape topology at the same time with some initialized shape. To alleviate this issue, we follow Nvdiffrecmc [17] and use soft occlusion during the first few hundred iterations. Specifically, the occluded light rays will be scaled by $\eta$, which gradually decreases from 1 at iteration 0 to 0 at iteration 1000. $\eta$ is fed to the denoiser in Nvdiffrecmc as well. Please refer to [17] for implementation details.

The learnable PBR materials are parameterized in the same way as in [17]. We do not learn a normal map (and therefore the normal regularization in the Nvdiffrecmc paper does not apply here).

We use Adam optimizer with $(\beta_1, \beta_2) = (0.9, 0.99)$. The total number of iterations for each shape is 5000. The initial learning rate of the MLP for parameterizing SDF values is set to $3e - 4$ while those for mSDF and grid node offsets are set to $0.15$. The initial learning rate for the rest of parameters used in **G-SHELL** with FlexiCubes are set to $3e - 4$. The learning rates follow a decay schedule of $10^{-0.0002 \cdot t}$ where $t$ is the iteration number.

We use an MLP with 6 hidden layers and a hidden channel size of 128 (with a concatenation shortcut from input to the output of the 4th layer) to parameterize SDFs. We use the standard positional encoding (for instance in [41]): $[1, \cos(2^1 x), \sin(2^1 x), \cos(2^2 x), \sin(2^2 x), ..., \cos(2^K x), \sin(2^K x)]$ with $K = 6$. We use Softplus activation with $\beta = 100$ for all layers except for the final output layer (in which no activation function is used). The MLP is initialized by fitting the SDF of a sphere with the diameter being roughly half of the grid size.

The learnable grid node offsets, after multiplied by the scale of deformation, are directly added to the canonical (*i.e.*, undeformed) grid vertex positions. After each gradient update, we clip the learned (and unscaled) grid node offsets to $[-1, 1]$. These learnable offsets are initialized to 0.

The images for the DeepFashion3D ground truth shapes are rendered using Cycles engine in Blender with 5,000 samples, a number of max bounces of 24 and a fixed environment lightmap.

We sample 100,000 points per mesh to compute Chamfer distances. Here we use the implementation in ECON [63] [8], which computes the bi-directional metric with pointcloud-to-surface distances.

---

[7]https://github.com/NVlabs/nvdiffrecmc
[8]https://github.com/YuliangXiu/ECON/blob/master/lib/dataset/Evaluator.py

## C    Detailed Settings on Diffusion Models with G-Shell

### C.1    Grid storage of interpolation coefficients from mSDF

As described in the main paper, in each tetrahedral cell there are 12 candidate (watertight) mesh edges if we count all possible SDF configurations. Since these 12 candidate edges are in general spatially separated, we take the simplest approach and store them on a larger cubic grid.

Suppose (normalized) SDF values and grid node offsets are stored in a cubic grid of resolution $R$, the same as in MeshDiffusion. Due to the spatial regularity of these 12 candidate edges (as long as we use a regular enough tetrahedral grid), we design a positional embedding for each edge. Specifically, the "location" of each candidate edge $(u_i, u_j)$, of which $u_i$ lies on the tetrahedral grid edge $(p_1, p_2)$ and $u_j$ on $(p_2, p_3)$, is

$$\frac{\overline{p}_1 + \overline{p}_3 + 2\overline{p}_2}{4} \tag{22}$$

where $\overline{p}$'s are the undeformed grid vertices respectively. With these positional embeddings, we may store them to the nearest coordinate in a larger cubic grid of resolution $2R$. As a result, we are able to use 3D U-Net for a diffusion model to jointly generate SDF values, deformation vectors and mSDF-induced interpolation coefficients.

For each candidate edge $(u_i, u_j)$, we may either generate $\alpha_i$ or $\alpha_j$ (note that $\alpha_j = 1 - \alpha_i$). We pick the simplest convention: we first sort two nodes $(u_i, u_j)$ of each candidate edge in a lexicographical order of the three dimensions of $a$ and $b$, and then for each sorted edge $(a, b)$, we always pick $\alpha_a$ as the interpolation coefficient to generate.

### C.2    Architecture and training settings

We use the same architecture of MeshDiffusion but add two input layers (one for the mSDF values and one for the mSDF grid mask which indicates where in the grid stores mSDF values; each implemented with a 3D convolution layer of kernel size 3) and an output layer (implemented with a transposed 3D convolution layer with kernel size 4, stride 2 and padding 1) to accommodate the introduction of an additional grid for mSDF-induced interpolation coefficients. The output of the additional input layer is directly added to the output of the original input layer in the MeshDiffusion U-Net architecture. Similar to MeshDiffusion, the predicted mSDF noise is multiplied by the mSDF grid mask so that only the used mSDF grid locations are counted.

We follow the training settings in MeshDiffusion with a reduced learning rate of $1e - 5$. We use a batch size of $8$ with $4$ gradient accumulation steps (with $8$ 80GB-mem A100 GPUs). We use mixed precision training to speed up the training process.

### C.3    Data preparation

We use the official GET3D implementation[9] to prepare datasets of rendered images for ground truth meshes (with a simple diffuse-only material). For MeshDiffusion (with DMTet), we mostly follow [36] but fit each object with $1000$ iterations for both coarse-fitting and finetuning stages (instead of $5000$) with the deformation scales set to $0$ and $2.0$, respectively. For G-MeshDiffusion, we use the depth supervision loss with $\gamma_{\text{depth}} = 100$ but disable the mSDF regularization losses as the depth information is enough for identifying topological holes on ground truth shapes. As in the reconstruction experiments, SDFs are parameterized by an MLP instead of being directly stored as learnable scalars in the tetrahedral grid. We recenter the meshes to the origin (in the world coordinate), and then rescale all meshes in an isotropic way so that the minimum and maximum of the bounding box coordinates are $-1$ and $1$, respectively. After rescaling, we compute the minimum and maximum of $x, y, z$ coordinates in each dataset (upper and lower garments) and scale the tetrahedral grid with $H = \left(0.45(x_{\max} - x_{\min}), 0.45(y_{\max} - y_{\min}), 0.45(z_{\max} - z_{\min})\right)$. The SDF values are initialized to fit a ellipsoid obtained by scaling a unit sphere with $H$. The mSDF values stored on tetrahedral grids are initialized in the same way as in reconstruction experiments: we randomly initialize them by sampling from a uniform distribution $\mathcal{U}(-0.01, 0.99)$.

---

[9]https://github.com/nv-tlabs/GET3D

# D  A BRIEF INTRODUCTION ON MESHDIFFUSION

MeshDiffusion [36] is a diffusion model [19] that generates a tetrahedral grid representation [55] for watertight meshes. It assumes that every mesh in the dataset is parameterized by a deformable tetrahedral grid (with fixed grid topology) which stores SDF values. By setting the canonical tetrahedral grid as a uniform grid, one may measure the node offsets from the deformed position to the canonical position. The deformable grid of SDF values (1 dimension) can therefore be turned into a uniform grid of offsets and SDF values (in total 4 dimensions). As a uniform tetrahedral grid can be seen as a subset of a cubic grid, a dataset of cubic grids can be created by introducing some artificial sites to augment the tetrahedral grid. By using cubic grids, standard 3D U-Nets (with standard 3D convolution layers) can be used for the diffusion model, and MeshDiffusion is trained on and produces such augmented grids.

While directly producing a 4-dimensional vector of node offsets and SDF values is feasible, often it produces many artifacts in generated meshes. The reason is mostly due to that the mesh vertex positions are computed in a non-linear way:

$$u_{12} = \frac{u_1 s_2 - u_2 s_1}{s_2 - s_1} = \frac{|s_2| u_1 + |s_1| u_2}{|s_1| + |s_2|} \tag{23}$$

with $u_1, u_2$ be two nodes of an edge on a tetrahedral grid and $s_1 > 0 > s_2$ the corresponding SDF values. Notice that $u'$ stays the same if one scales both $s_1$ and $s_2$ by a same positive scalar. As the SDF values are not directly computed from the ground truth shapes but instead optimized [36, 42, 55], they cannot be uniquely determined. As a result, the scale of them can vary in different positions and vary across the dataset, which makes the diffusion model harder to learn. To see why it could be an issue, let's suppose that there is a node in the tetrahedral grid which always has a negative SDF value with tiny scale in the dataset, and its surrounding nodes have positive SDF values with large scales. A small perturbation on the SDF values of the node with a tiny SDF scale may easily lead to a topological change in the extracted mesh, while one on the neighboring nodes is much less likely to change the mesh topology.

Here is another perspective to demonstrate the effect of the unconstrained scale: suppose the signs of SDFs are known (therefore, known mesh topology) and the tetrahedral grid is not deformable. The model now only needs to learn the scale of SDF values. Let's further assume a small and identical noise $\epsilon' > 0$ on all SDF values. We have

$$u_{12,\text{noisy}} - u_{12} = \frac{\epsilon'}{|s_1| + |s_2|}(u_2 - u_1) \tag{24}$$

If one trains a DDPM model [19] on the extracted mesh vertex positions instead (as the mesh topology is assumed to be known and fixed), the loss becomes

$$\mathbb{E}_{t \in [0,T], \epsilon \sim \mathcal{N}(0,I), U \in \mathcal{D}} w_t \|\epsilon - f_\theta(\alpha_t U + \sigma_t \epsilon, t)\|^2 \tag{25}$$

where $\mathcal{D}$ is the dataset of meshes, $U$ is the vector of all mesh vertex positions and $w_t$ is a weighting coefficient and $f_\theta$ is the denoising U-Net. With the mesh extraction formula with noisy SDF values, one may see that the standard Gaussian noise in mesh vertex positions in each local region is roughly the standard Gaussian noise in SDF values multiplied by the inverse of the average local SDF scale. Uneven SDF scales in the dataset lead to a unevenly weighted loss (in the sense of mesh vertex positions) on different data points and on different mesh vertex positions.

To alleviate this issue, MeshDiffusion employs a SDF normalization strategy which rounds SDF values in the dataset to $\pm 1$, depending on the sign of SDF values. Such an operation leads to additional errors, and therefore in the data collection process one needs to finetune the optimized grid node offsets after the normalization step. As the model is trained on normalized datasets, SDF values of the generated grids need to be normalized as well. Here we note that a similar phenomenon has been observed in [66] in which learning diffusion models to generated NeRF grid fields from an arbitrary NeRF dataset leads to artifacts and constraints on NeRF datasets have to be imposed for better quality.

# E  IMPLEMENTATION OF **G-SHELL**

---

**Algorithm 1** Mesh Extraction with **G-SHELL**

---

1: For any grid edge $(p_1, p_2)$ with opposite SDF signs, compute mesh vertex positions by $u = (p_1 s_2 - p_2 s_1)/(s_2 - s_1)$.
2: Project all mSDF values stored on the 3D grid to the extracted mesh vertices. The resulted mSDF value $\nu'$ for any $u$ extracted from $(p_1, p_2)$ is $\nu' = (\nu_1 s_2 - \nu_2 s_1)/(s_2 - s_1)$.
3: With the mSDF on watertight mesh vertices and SDF signs on grid nodes known, check the look-up table and determine the non-watertight mesh topology for each single cell. Compute the position of any boundary vertex with $o = (\nu_b u_a - \nu_a u_b)/(\nu_b - \nu_a)$ for any watertight mesh edge $(u_a, u_b)$ to be cut.

---

# F MORE QUALITATIVE EXAMPLES OF MULTI-VIEW MESH RECONSTRUCTION

Figure 11: Comparison between reconstruction w/ **G-SHELL** and baseline methods on instance 92 in DeepFashion3D dataset.

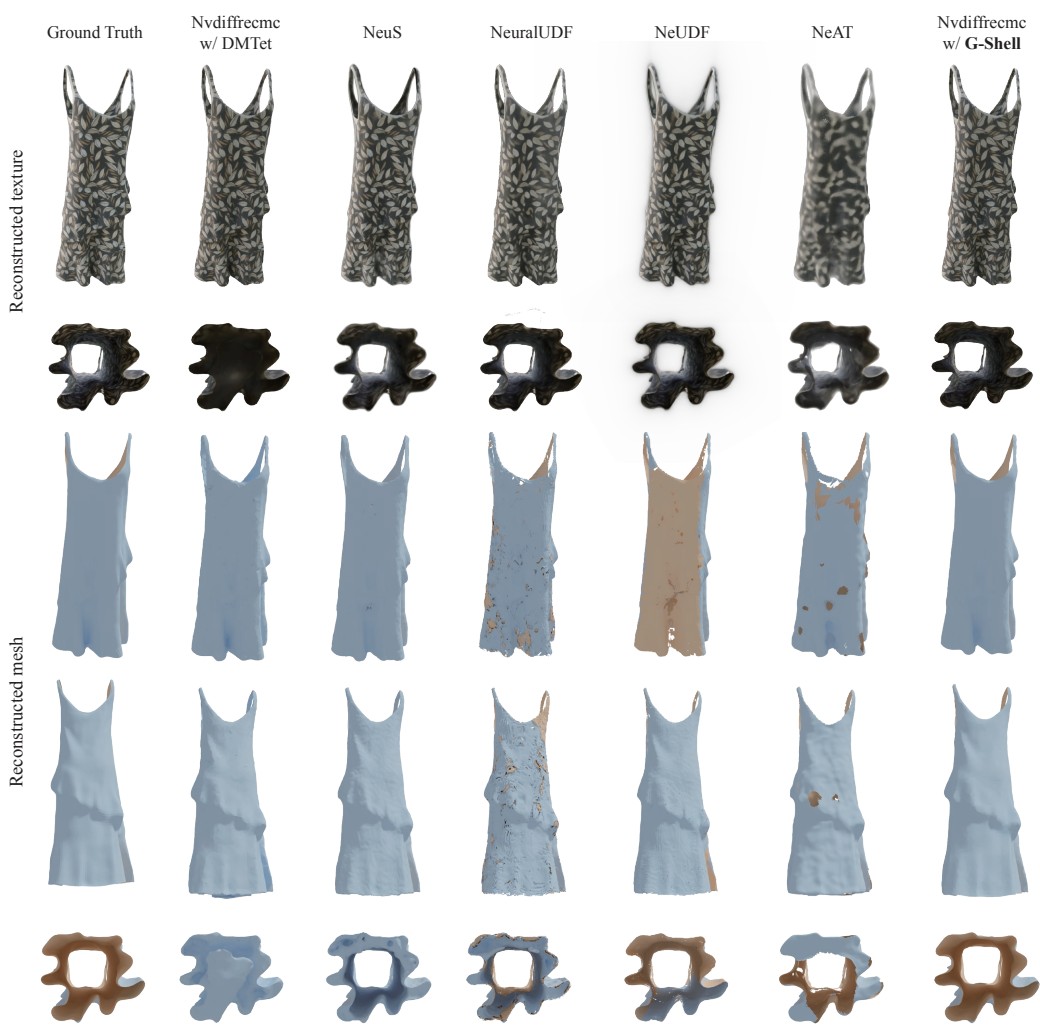

Figure 12: Comparison between reconstruction w/ **G-SHELL** and baseline methods on multiview reconstruction on instance 117 in DeepFashion3D dataset.

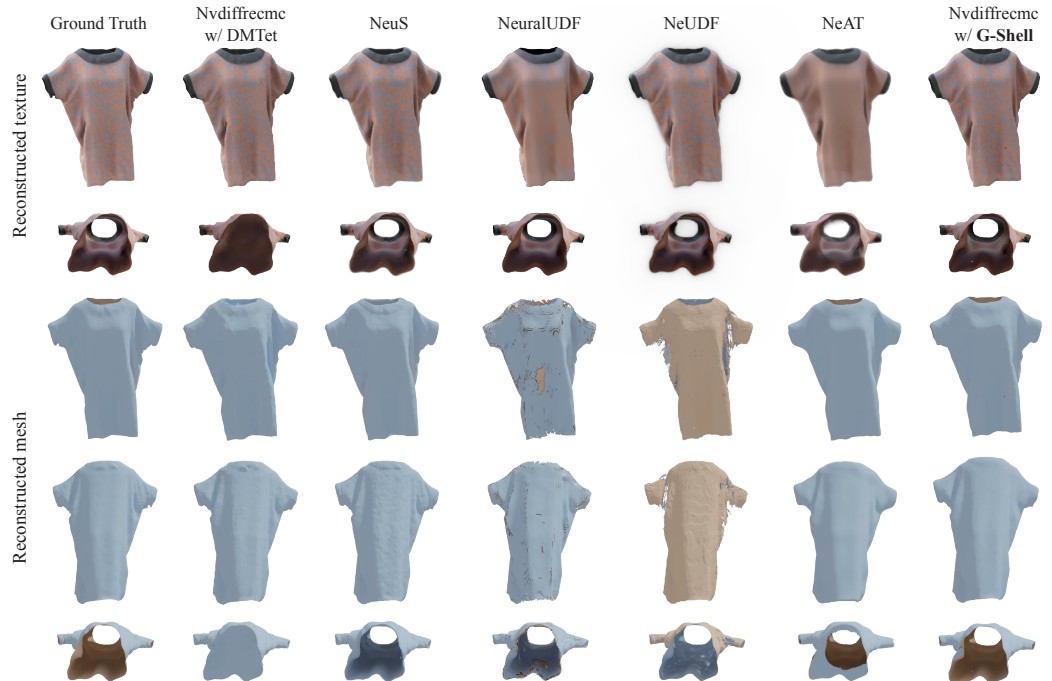

Figure 13: Comparison between reconstruction w/ **G-SHELL** and baseline methods on multiview reconstruction on instance 320 in DeepFashion3D dataset.

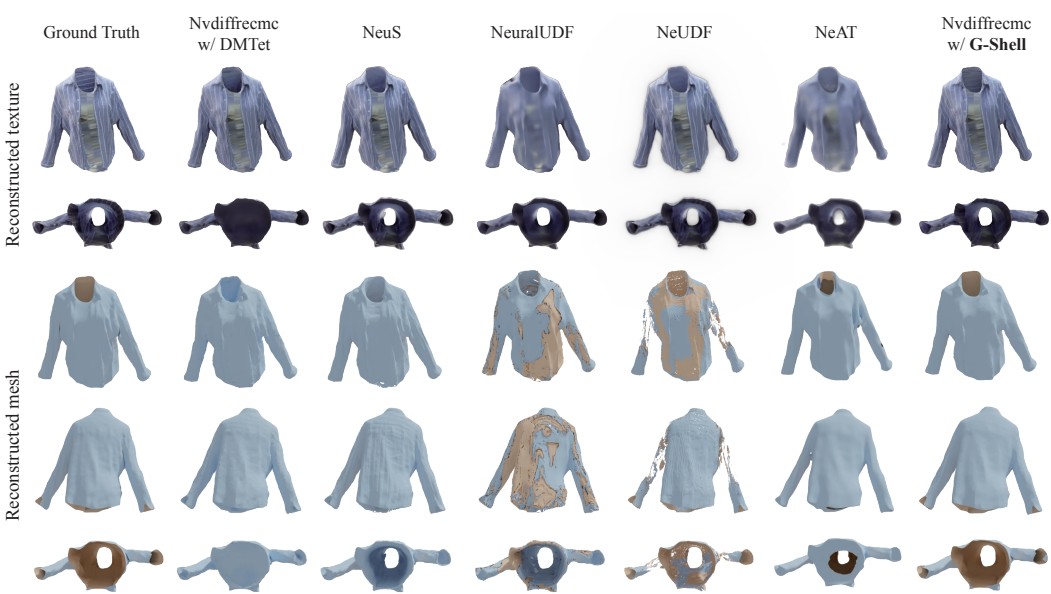

Figure 14: Comparison between reconstruction w/ **G-SHELL** and baseline methods on multiview reconstruction on instance 522 in DeepFashion3D dataset.

## G    RECONSTRUCTED MESHES FOR SURFACES WITH COMPLEX PATTERNS

We also show the reconstruction result of a non-watertight surface with complex surface patterns in Figure 15. The results show that **G-SHELL** can almost perfectly reconstruct the geometry.

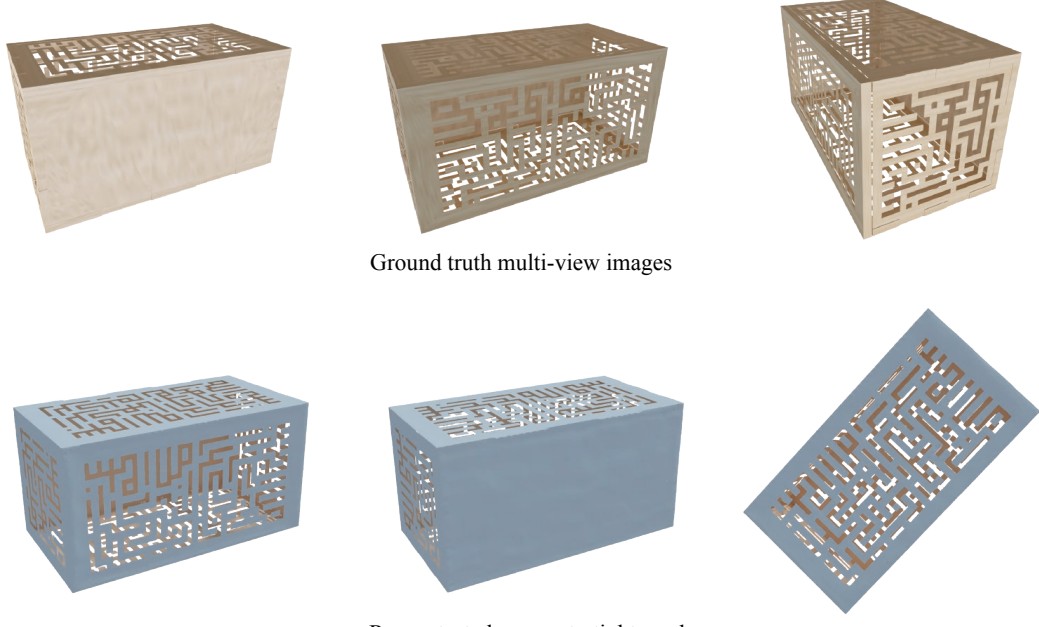

Ground truth multi-view images

Reconstruted non-watertight mesh

Figure 15: Reconstruction results of a non-watertight surface with complex surface patterns from multi-view imgaes.

# H    RECONSTRUCTION OF SHAPES WITH METALLIC MATERIALS

We visualize the ground truth and reconstructed shape in the specular lighting setting. With the increase of metalness in the material, the reconstructed shapes start to contain artifacts. Yet the reconstructed 3D meshes still replicate most of the target geometry and texture. The missing specular lighting effect in the rendered image is mostly due to the limitation of the renderer – no indirect illumination is considered. More advanced differentiable rendering methods may solve this issue.

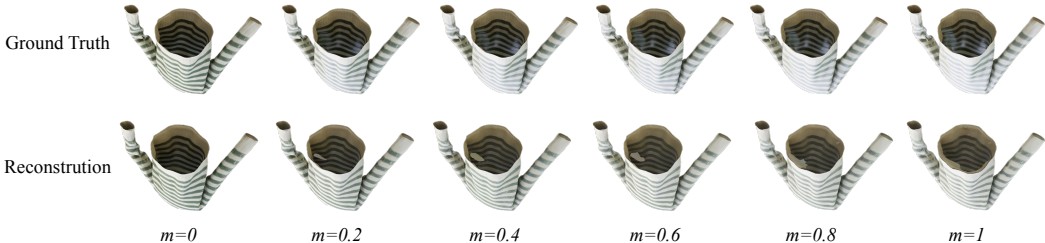

Figure 16: Qualitative results of the ablation study on the change of specular parameter in the ground truth mesh material. $m$ represents metalness parameter.

## I ON mSDF REGULARIZATION

It might be tempting to ignore the second term the mSDF regularization loss with some well-tuned set of loss coefficients. However, we empirically observe that it is hard to find a good choice for $\gamma_{\text{mSDF-reg-open}}$ to achieve conservative yet effective hole opening regularization. A too large $\gamma_{\text{mSDF-reg-open}}$ typically results in large topological holes when the shape to optimize is still far from the target shape (and often leads to completely empty meshes); a too small value fails to create topological holes when needed and fails to remove meshes in unobserved/occluded regions. It is possible to find a schedule of $\gamma_{\text{mSDF-reg-open}}$ without $L_{\text{mSDF-reg-close}}$. However, we note that the schedule of $\gamma_{\text{mSDF-reg-open}}$ has to be synchronized with the schedule of shadow ray contribution: once the occluded light rays hardly contribute in rendered colors, it is in general hard for the shape topology to change dramatically. A time-invariant loss coefficient of $L_{\text{mSDF-reg-close}}$ is in general easier to control and more robust.

On the other hand, a too small scale for $L_{\text{mSDF-reg-close}}$ often leads to incorrect shape topology while a too large scale for $L_{\text{mSDF-reg-close}}$ may produce some artifacts. These losses are correlated with different initialization strategies and influence the optimization process in a coupled way.

In Fig. 17, we show some extreme cases when inappropriate mSDF regularization scale leads to failures. We do not include the case for very large scales of $L_{\text{mSDF-reg-open}}$ as nearly no meshes will be produced.

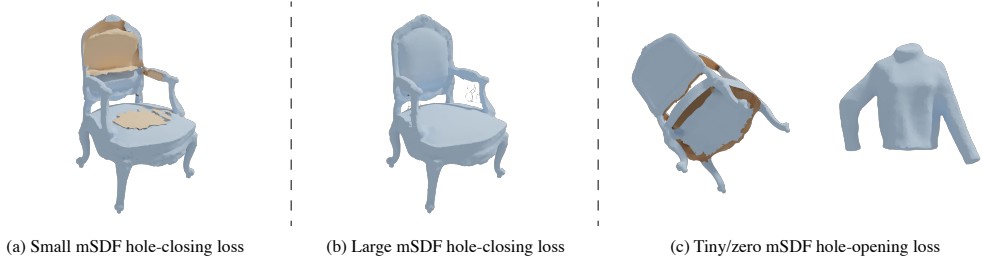

(a) Small mSDF hole-closing loss      (b) Large mSDF hole-closing loss      (c) Tiny/zero mSDF hole-opening loss

Figure 17: Extreme cases with inappropriate mSDF regularization scales.

# J    EXPERIMENT ON REAL DATA

To demonstrate that our method works on real data, we collected a set of multiview images by shooting a video with a hand-held smartphone. We extract 94 frames out of the video and obtain camera poses via COLMAP [51, 52]. The binary segmentation masks are obtained by running an off-the-shelf foreground segmentation model [47] on all the images. We fit the **G-SHELL** representation with the same set of parameters as in the chair example and show the results in Figure 18. Even the images include inconsistent lighting (due to the occlusion by the video shooter), motion blur and some specular lighting from the piece of the paper, Nvdiffrecmc inverse rendering with **G-SHELL** is still able to reconstruct a relatively reasonable shape with texture.

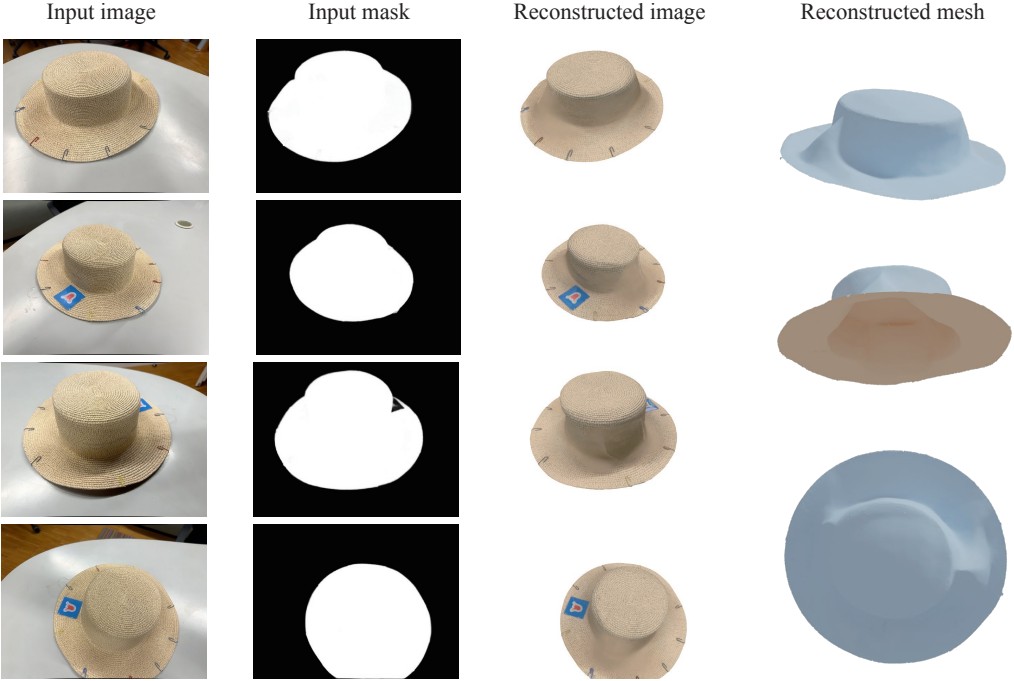

Figure 18: Empirical results on the real data collected by a hand-held smartphone.

## K   VISUALIZATION OF LEARNED WATERTIGHT MESH TEMPLATES

We show the resulted watertight mesh templates for some of the DeepFashion3D instances. We note that in theory these watertight mesh templates can be arbitrary, though the inductive bias of MLP and the Eikonal loss largely regularize the watertight template to be smooth enough.

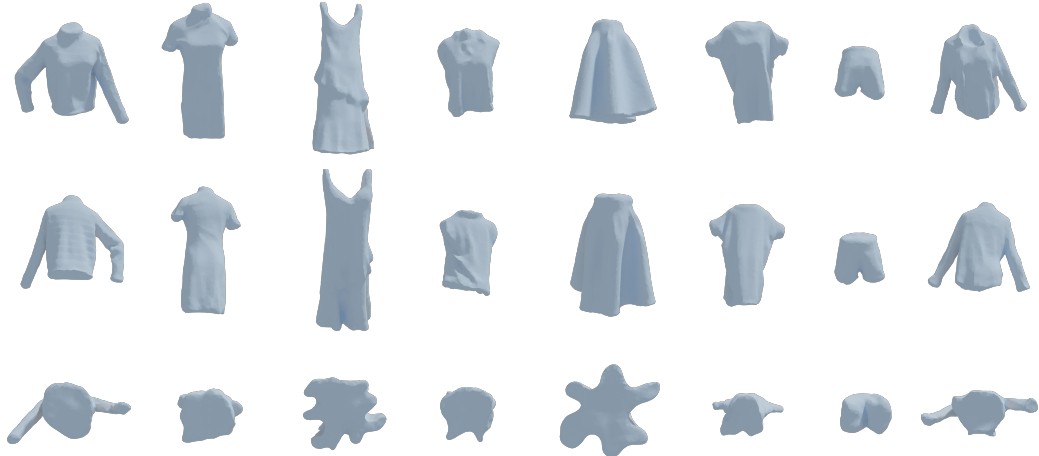

Figure 19: Visualization of the watertight mesh templates for each instance in DeepFashion3D dataset at the end of optimization.

