# OpenReview forum: "Ghost on the Shell: An Expressive Representation of General 3D Shapes"
_ICLR.cc/2024/Conference — ICLR 2024 oral_

### Official Review · Reviewer_9iXa · 2023-10-31

**Soundness:** 3 good
**Presentation:** 4 excellent
**Contribution:** 3 good
**Rating:** 8
**Confidence:** 3

**Summary:**

This paper presents a novel mesh representation called Ghost-on-the-Shell (G-SHELL). This representation parametrizes non-watertight surfaces by defining a manifold signed distance field on watertight templates. It enables reconstruction from multiview images and generative modeling of both watertight and non-watertight meshes of arbitrary topology. The paper demonstrates that G-SHELL performs well in tasks related to non-watertight mesh reconstruction and generation while also being effective for watertight meshes.

**Strengths:**

The paper introduces an original approach to implicitly modeling non-watertight 3D meshes. Treating open surfaces as entities floating on watertight surfaces is a novel idea with significant advantages over other methods that require unsigned distance fields (UDF). It leads to the development of a manifold signed distance field (mSDF) on the watertight template, which is a sound contribution.

In terms of quality, the methodology is well-described, and the authors offer a clear rationale for their Ghost-on-the-Shell (G-SHELL) implicit modelling approach. The paper also provides empirical evidence of the advantages of G-SHELL in tasks such as mesh reconstruction and generation, which enhances the overall quality of the work.

The paper is well-written and the contributions are cleary presented. The authors effectively communicate complex concepts.

In terms of significance, G-SHELL is a differentiable and efficient implicit representation for both watertight and non-watertight meshes which broadens its impact. Besides it has the potential to be easily adopted in computer graphics pipelines.  G-SHELL applications in reconstruction and mesh generation problems are notable. In particular, G-SHELL’s enables reconstruction methods where the topology, material and lighting are jointly optimized which is crucial for scaning 3D assets from images.

Overall, this paper presents an original, high-quality, clear, and significant contribution to deal with non-watertight surfaces and demonstrating practical applicability through empirical validation.

**Weaknesses:**

G-SHELL inherits common disadvantages of implicit surface representation methods. Mainly, since it employs a regular grid, surfaces with high and unbalanced entropy will require many grid elements, which can be inefficient and limiting for real applications.

G-SHELL uses a marching cubes-like algorithm to extract the surface. Even though this method is highly parallelizable, it also represents a burden in computation compared to explicit methods.

G-SHELL does not model self-intersecting and non-orientable surfaces.

**Questions:**

Regarding the experiments in mesh generation, G-SHELL is compared against other implicit  SDF based methods that can only represent watertight surfaces. Since the experiments are only conducted with a dataset of non-watertight surfaces (clothes), it is thus expected that G-SHELL is going to be better at the job than the rest of methods. How well MD w/ G-SHELL performs with watertight surfaces compared with MeshDiffusion?


PolyGen [40] is regarded on page 3 as ineffective for high vertex counts. Although this is probably accurate, it is worth clarifying in the paper that methods like PolyGen generate non-uniform meshes and use the vertices more efficiently than the regular grid used in G-SHELL.  Comparing vertex counts between the two methods does not seem fair.


The paper lacks experiments on real data, especially in reconstruction. It would be interesting to compare G-SHELL + differentiable rasterizers against the baselines based on volume rendering or even vanilla NERF + marching cubes. Do the authors plan to include such experiments in the camera-ready version?


It would be illustrative to visualize the watertight template being estimated jointly with the open surface in some of the experiments.

---

> ### Author Response · Authors · 2023-11-16
> **Response to Reviewer 9iXa**
>
> We thank the reviewer for their time and efforts in reviewing our paper, their acknowledgement of contribution and their suggestions.
>
> > G-SHELL inherits common disadvantages of implicit surface representation methods. Mainly, since it employs a regular grid, surfaces with high and unbalanced entropy will require many grid elements, which can be inefficient and limiting for real applications.
>
> We totally agree with the reviewer that grid-based representations can be inefficient for modeling many complex geometries. Nevertheless we would like to point out that, for many scenarios, it has been demonstrated that the grid-based representations can behave reasonably well (e.g. Fig. 13 and 14 in the Nvdiffrec paper [1]) with a single GPU.
>
> > Regarding the experiments in mesh generation, G-SHELL is compared against other implicit SDF based methods that can only represent watertight surfaces. Since the experiments are only conducted with a dataset of non-watertight surfaces (clothes), it is thus expected that G-SHELL is going to be better at the job than the rest of methods. How well MD w/ G-SHELL performs with watertight surfaces compared with MeshDiffusion?
>
> Thank you for bringing up this point. To clarify: G-MeshDiffusion (“MD w/ G-Shell” in the first edition) is a superset of MD. By setting mSDF values to +1, G-MD reduces to the original MD model. As a result, the performance of G-MD on watertight mesh generation is basically the same as that of MeshDiffusion.
>
> > PolyGen [40] is regarded on page 3 as ineffective for high vertex counts. Although this is probably accurate, it is worth clarifying in the paper that methods like PolyGen generate non-uniform meshes and use the vertices more efficiently than the regular grid used in G-SHELL. Comparing vertex counts between the two methods does not seem fair.
>
>
> Thank you for pointing out this aspect. We significantly polished the related work section to avoid confusion.
>
> We would like to further clarify some difficulties faced in PolyGen. First, it is relatively hard for autoregressive models to have a long (and varying-length) sequence, which is evidenced by the literature in transformers and language models. Besides, there is no guarantee on manifoldness (to be more precise, the guarantee on being “manifolds with boundary”) and PolyGen easily creates very long non-manifold edges when the sequences in the dataset become more complex. In addition, even for the same shape, the mesh topology can vary a lot (different resolutions, different remeshing methods, etc.) and it is generally hard for an autoregressive model to handle such cases. For meshes that are manually created by designers for regular shapes like IKEA furniture, PolyGen may have an advantage over grid-based methods.
>
>
> > The paper lacks experiments on real data, especially in reconstruction. It would be interesting to compare G-SHELL + differentiable rasterizers against the baselines based on volume rendering or even vanilla NERF + marching cubes. Do the authors plan to include such experiments in the camera-ready version?
>
>
> For generation, we would like to point out that there is a lack of real datasets for non-watertight meshes, mostly due to the difficulties in mass-reconstruction of 3D non-watertight meshes. Existing large datasets for non-watertight meshes are mostly for clothing / clothed humans (e.g. CLOTH3D [2], BEDLAM [3]) and all of them are synthetic. Even for these datasets, the diversity of clothing styles is limited. As our purpose is only to explore the representation, we do not explore building a better dataset for non-watertight meshes.
>
> For reconstruction, we plan to include results of our method and baselines on real data. Due to time and resource constraints, for now we only show in the appendix the results of our method on data collected from a smartphone — with our method, the shape can be roughly reconstructed even under inconsistent lighting (due to occlusion during video shooting) and motion blur. In the meantime, we would like to emphasize that our method is primarily concerned with geometry - those volumetric-based methods typically face difficulties due to the flexibility of open surfaces (compared to closed ones).
>
>
> > It would be illustrative to visualize the watertight template being estimated jointly with the open surface in some of the experiments.
>
>
> Thank you for your valuable suggestion. We have included the learned watertight template in the appendix of the revised draft. In the meantime we would like to emphasize that, for most of the experiments, no loss is computed on the discarded regions on the watertight template and therefore the shape of the watertight template can be arbitrary (though regularized to some degree by the Eikonal loss on SDF values).

---

> > ### Author Response · Authors · 2023-11-16
> > **Citations for the Response**
> >
> > ---
> > [1] Extracting Triangular 3D Models, Materials, and Lighting From Images. Jacob Munkberg, Jon Hasselgren, Tianchang Shen, Jun Gao, Wenzheng Chen, Alex Evans, Thomas Müller, Sanja Fidler. CVPR 2022.
> >
> >
> > [2] CLOTH3D: Clothed 3D Humans. Hugo Bertiche, Meysam Madadi, Sergio Escalera. ECCV 2020.
> >
> > [3] BEDLAM: A Synthetic Dataset of Bodies Exhibiting Detailed Lifelike Animated Motion. Michael J. Black, Priyanka Patel, Joachim Tesch, Jinlong Yang. CVPR 2023.

---

> > ### Comment · Reviewer_9iXa · 2023-11-23
> > **Thank you for the clarification**
> >
> > The authors have addressed my questions satisfactorily and made considerable effort in the rebuttal, adding new experiments with real data. In addition, given the responses to the rest of the reviewers, I will keep my very positive rating of the paper.

---

### Official Review · Reviewer_dsfR · 2023-11-01

**Soundness:** 4 excellent
**Presentation:** 3 good
**Contribution:** 3 good
**Rating:** 8
**Confidence:** 5

**Summary:**

This paper extends the 3D grid representation by introducing an additional mSDF field defined on grid vertices for the modeling of open surfaces.  This new representation can be used in the reconstruction and generative modeling of open 3D surface meshes.

**Strengths:**

1. A nice extension of 3D grid representation that can handle open surface meshes.
2. Impressive experimental results on open surface reconstruction and generation.

**Weaknesses:**

While the experimental results are impressive, it is not clear whether the proposed modified marching cube or tetrahedra with mSDF value can guarantee the correct topology of the 3D mesh. For example, an isolated edge with no incident triangles.

**Questions:**

1. The ability to handle specular surfaces is emphasized in the experimental results. However, the experimental setting to handle the specular surfaces is not clear enough.  Do you simultaneously reconstruct lighting, geometry and material properties from the captured images?  or you assume lighting condition is known.

2. How is the performance of the proposed method when reconstructing open surfaces that are homeomorphic to donuts or other complex surfaces with many topology handles?

---

> ### Author Response · Authors · 2023-11-16
>
> We appreciate the reviewer for acknowledging our contribution and raising their concerns.
>
> > While the experimental results are impressive, it is not clear whether the proposed modified marching cube or tetrahedra with mSDF value can guarantee the correct topology of the 3D mesh. For example, an isolated edge with no incident triangles.
>
> We apologize for not explaining such an important aspect in detail.
>
> Let us first consider the case where we use a uniform and non-deformable grid to discretize the space. Indeed, a key benefit of using SDF is that no isolated edges will be produced as an SDF always produces manifold meshes. Similarly, mesh extraction with mSDF cuts a watertight mesh without breaking the (1-dimensional) manifoldness on 2-manifolds. The final mesh will therefore contain no isolated edges. Another way to understand the no-self-intersection property is to treat the “cutting” operation as an intersection between the watertight template and a “virtual watertight volume” defined by the mSDF field (before being projected onto the watertight template).
>
> For the deformable grid case: as long as we constrain the node offsets in the tetrahedral/cubic grid to some reasonable range (less than the half of the shortest edge in the initial grid will be sufficient), it is guaranteed that there will not be self-intersecting faces.
>
> There are cases where one may want to use larger-scale offsets (for instance to capture finer geometric details without paying an additional price in GPU memory) or some techniques that break the manifoldness assumption (for example FlexiCubes [1]). In such cases, one can use standard Laplacian smoothing as a post-processing step, or introduce some mesh regularity loss (e.g. the mean angle deviation loss in [1]; related results in Fig. 12 in their main paper and Fig. 2 in their appendix).
>
> > The ability to handle specular surfaces is emphasized in the experimental results. However, the experimental setting to handle the specular surfaces is not clear enough. Do you simultaneously reconstruct lighting, geometry and material properties from the captured images? or you assume lighting condition is known.
>
> Yes. We follow the Nvdiffrecmc paper [2] and simultaneously optimize geometry (SDF, mSDF, offsets), lighting and material.
>
>
> > How is the performance of the proposed method when reconstructing open surfaces that are homeomorphic to donuts or other complex surfaces with many topology handles?
>
> A: Good question! To demonstrate the capability of the proposed inverse rendering method, we have included an example in the appendix with a more complex pattern on the surface, where there are irregularly-shaped topological holes and isolated connected components.
>
> Another example is the chair experiment shown in Sec 6.2. We have revised the visualization for the reconstructed chair (where the input views in the dataset only view the ground truth chair from the above), and it is probably now easier to see the topological feature of the reconstructed shape — the reconstructed chair legs are “watertight” similar to the handles of teapots.
>
> ---
>
> [1] Flexible Isosurface Extraction for Gradient-Based Mesh Optimization. Tianchang Shen, Jacob Munkberg, Jon Hasselgren, Kangxue Yin, Zian Wang, Wenzheng Chen, Zan Gojcic, Sanja Fidler, Nicholas Sharp, Jun Gao. SIGGRAPH 2023.
>
> [2] Shape, Light, and Material Decomposition from Images using Monte Carlo Rendering and Denoising. Jon Hasselgren, Nikolai Hofmann, Jacob Munkberg. NeurIPS 2022.

---

> > ### Comment · Reviewer_dsfR · 2023-11-18
> >
> > Thanks for the detailed response to my questions.   My concerns are addressed.
> >
> > I will keep my score in the final evaluation.

---

### Official Review · Reviewer_hJqu · 2023-11-03

**Soundness:** 3 good
**Presentation:** 2 fair
**Contribution:** 3 good
**Rating:** 8
**Confidence:** 4

**Summary:**

The paper proposes a new representation for surfaces with boundary (open surfaces) geared toward inverse rendering and surface reconstruction. The representation views an open surface as a sub-level set of a function on a closed surface, which is in turn represented as a level set of a function in the ambient space. The two functions are discretized together on a background grid, and an extended marching-tetrahedra lookup table enables extraction of a mesh for the open surface. The advantages of the representation are demonstrated for reconstruction from images as well as generative modeling by a diffusion method.

**Strengths:**

The basic idea is elegant, and it avoids the problems of unsigned distance fields. The comparisons to previous work are also compelling. The generative modeling results look cool, though it would be nice to see some examples other than clothing if other datasets are available.

**Weaknesses:**

The authors claim this is the first work to propose a differentiable representation suitable for both open and closed surfaces. Though they mention representations for open surfaces based on unsigned distance fields, they should also include citations to the following two works, which offer alternative approaches:
- D. Palmer, D. Smirnov, S. Wang, A. Chern, and J. Solomon, “DeepCurrents: learning implicit representations of shapes with boundaries,” in Proceedings of the IEEE/CVF Conference on Computer Vision and Pattern Recognition, 2022, pp. 18665–18675.
- T. V. Christiansen, J. A. Bærentzen, R. R. Paulsen, and M. R. Hannemose, “Neural Representation of Open Surfaces,” 2023, doi: 10.1111/cgf.14916.

The exposition could use some polishing. In addition to general copy-editing, some details of the method require further elaboration. Most prominently, the mesh extraction algorithm, which is a main contribution of the paper, is described only very briefly in section 4.2, and the lookup table for G-shells is only explained pictorially in figure 3. It would be helpful to readers to include at least a little more explanation of what is going on in that figure and why (even if it has to go in an appendix or supplemental material).

The description of the generative modeling approach in the paragraphs following eq. (2) is unclear. What does "unevenly weighted prediction" mean? What is the naïve diffusion loss, and what are you replacing it with? Why does predicting the linear interpolation coefficient instead of the value of $\nu$ help? Is there any tradeoff in doing this? If the normalization of SDF values is an issue, would there be an advantage to using more general implicit functions instead? If you are going to extend MeshDiffusion, it would be helpful to include at least a brief summary of how that method works.

**Questions:**

The paper shows a lookup table for a marching-tetrahedra approach for G-shells. Is there also an extended version of marching-cubes?

The "hole-opening" loss in eq. (1) sounds like it really promotes hole-closing. In any case, it requires more explanation. What does the parameter $\epsilon$ do? How should one set it? The paper refers to "topological holes" but I assume what is meant here are boundaries, not handles.

The appendix lists many more loss terms with free parameters, and it is unclear whether all of these terms are necessary. As usual, it would be helpful to add an ablation study. Some of the descriptions of the loss terms are cryptic and could do with elaboration. E.g., how does (12) reduce "floaters" and "inner geometry"? Why do you need a second SDF regularization after imposing the eikonal regularization? Would it be helpful to include some form of eikonal regularization on $\nu$ as well—ideally taking the gradient along the surface?

## Minor quibbles:
- p. 2: "generation modeling" -> "generative modeling"
- p. 3: What does "validity value" mean?
- p. 4: referring to a curve as a "2D mesh" and a surface as a "3D mesh" is confusing. It would be better to say "polygonal curve" and "surface mesh," respectively, or something similar.
- p. 5: "pose-processing" -> "post-processing"
- 6.2: "winding number" should really be "generalized winding number." Also, I am not sure what it means to look at the generalized winding number on the surface when it is a function in the ambient volume that has a sharp jump at the surface.

---

> ### Author Response · Authors · 2023-11-16
> **Response to Reviewer hJqu**
>
> We appreciate the reviewer for acknowledging our contribution and proposing valuable suggestions. We have corrected the typos and inaccurate descriptions pointed out by the reviewer.
>
> > Cite two papers.
>
> We thank the reviewer for pointing out these two relevant papers. We have included them in the revised draft.
>
> > The exposition could use some polishing. In addition to general copy-editing, some details of the method require further elaboration. Most prominently, the mesh extraction algorithm, which is a main contribution of the paper, is described only very briefly in section 4.2, and the lookup table for G-shells is only explained pictorially in figure 3. It would be helpful to readers to include at least a little more explanation of what is going on in that figure and why (even if it has to go in an appendix or supplemental material).
>
> Thank you for your suggestion. We have included in the revised draft an algorithm box on the mesh extraction algorithm in Appendix E, plus Fig. 3 and some lines giving a more informative explanation in the main paper.
>
>
> > The description of the generative modeling approach in the paragraphs following eq. (2) is unclear. What does "unevenly weighted prediction" mean? What is the naïve diffusion loss, and what are you replacing it with? Why does predicting the linear interpolation coefficient instead of the value of help? Is there any tradeoff in doing this? If the normalization of SDF values is an issue, would there be an advantage to using more general implicit functions instead? If you are going to extend MeshDiffusion, it would be helpful to readers to include at least a little more explanation of what is going on in that figure and why (even if it has to go in an appendix or supplemental material).
>
> We apologize for not clearly explaining the details. To further clarify the underlying logic, we have detailed in the revised draft the rationale behind the SDF normalization process in MeshDiffusion. For the ease of reviewing (especially for other reviewers), please let us repeat the idea below:
>
> The naive diffusion loss means simply building a dataset with no restriction on SDF/mSDF values and using the standard L2 diffusion loss to train. The problem is that the learned SDF values are only values up to a scale (locally) of the ground truth ones —- for some regions the SDF scales can be huge while for some other regions the values can be tiny. As a hypothetical case: if we already know the SDF signs but not the scales, the uneven SDF scale across different regions leads to different sensitivity of interpolated (watertight) mesh vertex positions to the SDF values. The resulting model, with such an unprocessed dataset, will not perform well (similar phenomena have been observed in papers such as [1], where learning a good diffusion model with unprocessed datasets is extremely hard).
>
> By predicting the linear interpolation coefficients, the effect of the imbalance in the scale of SDF values is removed (since the scale on the numerator and denominator of the marching tetrahedra formula cancel each other). The trade-off is mostly on the demand for memory —- one has to store and infer more information at a finer granularity.
>
> The need for SDF normalization is mostly due to the parametrization with Marching Cubes/Tetrahedra, in which the SDF values are not obtained in a controllable way and the mesh vertex positions are computed as a nonlinear function of SDF values. At this point we are not aware of more efficient and effective mesh parameterization (e.g., alternative methods like Neural Dual Contouring [2] do not work as shown in Fig. 4 of [3]).
>
> Finally, our intuition is that, for any implicit function without a simple and robust enough mesh extraction rule (or a huge dataset + model capacity), it can be hard for diffusion models to generalize well on that representation unless we do some sort of “pre-processing” — training a VAE for latent features as in [2], dataset normalization as in our paper and [1], etc.
>
>
>
> > The paper shows a lookup table for a marching-tetrahedra approach for G-shells. Is there also an extended version of marching-cubes?
>
> Thank you for your suggestion, and we feel that an illustration for the cubic-case look-up table would be very helpful. However, drawing them is probably a bit demanding given the short period of time since there are dozens of  possible cases if we manually enumerate all of them even after removing all symmetry cases. We briefly explain how it is constructed. Take any case in the Marching-Cubes look-up table (Figure in https://en.wikipedia.org/wiki/Marching_cubes). For any (planar) polygon, we can enumerate all possible ways to color the vertices (positive / negative) and therefore all possible ways to cut the polygon. For multiple (planar) polygons in the same cell, the numbers of cutting types for individual polygons are multiplied.

---

> ### Author Response · Authors · 2023-11-16
> **Response to Reviewer hJqu (Continued, 1/2)**
>
> > The "hole-opening" loss in eq. (1) sounds like it really promotes hole-closing. In any case, it requires more explanation. What does the parameter `epsilon` do? How should one set it? The paper refers to "topological holes" but I assume what is meant here are boundaries, not handles.
>
>
> We have rewritten the loss in the revised draft in a more mathematically sound and clearer way. The first term indeed encourages hole-opening. The second term, which is computed only on the boundary vertices (therefore of mSDF values 0), encourages hole-closing because we are moving the mSDF value from “$\nu$ minus a positive $\epsilon$“ to “zero”. The epsilon behaves like the margin in a Hinge loss (except that we are using an $L_2$ instead of $L_1$ loss). We simply set it to an arbitrary moderately-positive scalar (not greater than 1 for numerical stability) and leave the control of the loss scale to the multiplier on the loss term.
>
> We give an intuitive explanation for using the term “topological holes” with an example: without the hole-opening regularization loss, the holes at the lower ends of sleeves, in the T-shirt instance shown in the paper, can be completely closed due to initialization. Adding the regularization will encourage the mSDF values there to be negative. Once some values become negative, holes appear at originally closed regions. The boundaries of these holes now are free to contract or expand, depending on the image signal as well as the binary mask signal.
>
> We have included additional qualitative results on the mSDF regularization in the appendix to demonstrate the effect of the two terms. A short summary of the results: 1) a large hole opening loss leads to optimization failures, while a tiny hole opening loss fails to create holes when initialization is bad, plus that the meshes in the never-observed regions fail to be removed; 2) a too small hole closing loss leads to incorrectly optimized shapes, while a too large hole closing loss may lead to artifacts on the boundary.
>
> > The appendix lists many more loss terms with free parameters, and it is unclear whether all of these terms are necessary. As usual, it would be helpful to add an ablation study. Some of the descriptions of the loss terms are cryptic and could do with elaboration.
>
> Thank you for raising this important issue, and we totally agree with the reviewer that the amount of free hyperparameters is more than in some other tasks. We provide some qualitative results on extreme cases of mSDF regularization to validate the necessity of this specific loss, as it is not present in other related papers (to our knowledge). However, it would be computationally demanding to perform a full batch of ablation study on other hyperparameters.
>
>
> All the other regularization losses (except for the Eikonal loss) are directly borrowed from the differentiable mesh renderer papers (Nvdiffrec [3], Nvdiffrecmc [4]) —- they are mostly specific to the renderers but not our representation. These papers prove that their regularization losses are necessary for learning the watertight meshes with “differentiable rasterization + Marching-Cubes-like methods”. As we parameterize non-watertight meshes as floating on learned watertight mesh templates, these regularization losses are helpful for excluding artifacts related to watertight mesh templates. Indeed, one of our objectives is to propose a unified representation for both watertight and non-watertight meshes. To cover the pure watertight cases, these regularization losses are necessary (since mSDF values are basically all positive on the watertight mesh template).
>
> We apologize for not clearly explaining the regularization losses in detail, and we have added some more details in describing the losses in the revised draft to make our paper more readable.
>
> If the current level of detail in the revised draft is not enough, we would like to refer interested readers to the original papers. Fully describing them will probably require several pages, especially given that we have included experiments with “G-Shell + FlexiCubes” in which it is almost impossible to explain the mesh regularization without fully explaining their method. And we feel like that the differentiable renderer is not our contribution — to some degree, we are merely using them (with necessarily adaptations) for non-watertight mesh cases.
>
> > how does (12) reduce "floaters" and "inner geometry"?
>
> The SDF regularization, introduced in Nvdiffrec, penalizes the SDF values in the unobserved regions (corresponding to the “inner geometry”) and encourages the SDF to be smooth enough in the 3D space (so fewer “floaters” in the empty but observable space).

---

> ### Author Response · Authors · 2023-11-16
> **Response to Reviewer hJqu (Continued, 2/2)**
>
> > Why do you need a second SDF regularization after imposing the eikonal regularization?
>
> Good question! The Eikonal loss does not necessarily remove floaters since they are computed only on the surface but not the whole 3D space. Due to the same reason, this SDF regularization loss does not help too much on learning a smooth but complex surface.
>
> > Would it be helpful to include some form of eikonal regularization on `nu` as well—ideally taking the gradient along the surface?
>
> Good question! We indeed tried to parameterize $\nu$ with an MLP but eventually found that the hyperparameters can be hard to tune (resulting meshes often disappear at some point in the training process). Therefore, we parameterize $\nu$’s explicitly on grids without an MLP. An Eikonal loss with explicitly-stored SDF values may be possible by using finite differences but we feel that it is beyond the scope of our paper. It would definitely be worth exploring these topics in the future.
>
>
> > p. 3: What does "validity value" mean?
>
> “Validity value” in the context of NeAT means the value used to determine if one cell should be considered or not by Marching Cubes in the post-processing stage.
>
>
> > p. 4: referring to a curve as a "2D mesh" and a surface as a "3D mesh" is confusing. It would be better to say "polygonal curve" and "surface mesh," respectively, or something similar.
>
> By “2D mesh”, we would like to draw an analogy between isoline extraction on 2-manifolds and isosurface extraction in the Euclidean 3D space. We have added additional explanations in the revised draft to avoid confusion.
>
>
> > 6.2: "winding number" should really be "generalized winding number." Also, I am not sure what it means to look at the generalized winding number on the surface when it is a function in the ambient volume that has a sharp jump at the surface.
>
> Thank you for correcting us – yes, what we computed is the generalized one. We are inspired by Figure 6 in [5] — the degree of “surface openness” is highly related to the winding numbers (red color intensity).
>
> On the discontinuity issue: we greatly thank the reviewer for pointing out our omission of this important technical issue. We now visualize the generalized winding number of the surrounding point clouds instead of on the surface.
>
> In addition, we now visualize the front and back faces with different colors — so it is probably easier to see how the chair is reconstructed with images when only viewing the chair from above.
>
> ---
>
> [1] Learning a Diffusion Prior for NeRFs. Guandao Yang, Abhijit Kundu, Leonidas J. Guibas, Jonathan T. Barron, Ben Poole. Arxiv 2304.14473.
>
> [2] 3D Neural Field Generation using Triplane Diffusion. J. Ryan Shue, Eric Ryan Chan, Ryan Po, Zachary Ankner, Jiajun Wu, Gordon Wetzstein. CVPR 2023.
>
> [3] Extracting Triangular 3D Models, Materials, and Lighting From Images. Jacob Munkberg, Jon Hasselgren, Tianchang Shen, Jun Gao, Wenzheng Chen, Alex Evans, Thomas Müller, Sanja Fidler. CVPR 2022.
>
> [4] Shape, Light, and Material Decomposition from Images using Monte Carlo Rendering and Denoising. Jon Hasselgren, Nikolai Hofmann, Jacob Munkberg. NeurIPS 2022.
>
> [5] Jacobson, Alec, Ladislav Kavan, and Olga Sorkine-Hornung. "Robust inside-outside segmentation using generalized winding numbers." ACM Transactions on Graphics (TOG) 32.4 (2013): 1-12.

---

### Official Review · Reviewer_RYwJ · 2023-11-04

**Soundness:** 4 excellent
**Presentation:** 4 excellent
**Contribution:** 3 good
**Rating:** 5
**Confidence:** 5

**Summary:**

This paper proposes a new representation, G-Shell, for 3D data. Existing papers mostly focus on modeling solid, watertight shapes, while the modeling of thin, open surfaces has not been widely studied. To fill this gap, this paper proposes a parameterization and develops a gruid-based method for both watertight and non-watertight meshes of arbitrary topology. Experiments show that G-Shell achieves state-of-the-art performance on non-watertight mesh reconstruction and generation tasks, while achieving competitive results for watertight meshes.

**Strengths:**

1. The topic studied in this paper is interesting and important.
2. Being able to model thin, open surfaces will be useful for a couple of applications.
3. The proposed method is technically sound.
4. Experiments show that the proposed method is effective in modeling non-watertight meshes.

**Weaknesses:**

1. While the proposed method is faster than other methods as shown in Table 3, 3 hours is still too long.
2. The thin shape examples in Figure 4 and Figure 7 don't have complicated geometry. If there is a more complicated geometry, how well would the proposed method perform in reconstructing / modeling? For instance, when dropping a cloth onto an object, the cloth will have a lot of folds, wrinkles and even a lot of self-contacts. Can the proposed method deal with this case?

**Questions:**

Please see questions above

---

> ### Author Response · Authors · 2023-11-16
>
> We thank the reviewer for acknowledging our contribution and raising the questions.
>
> > While the proposed method is faster than other methods as shown in Table 3, 3 hours is still too long.
>
> Thank you for raising your concern. In this paper, we would like to focus on the representation itself but not the detailed downstream tasks (reconstruction, generation, etc.). We position our paper as a concept-level proof that G-Shell, as a new representation, has the potential to enable fast and robust modeling of non-watertight meshes — similar to the position of NeRF with respect to the improved methods in the following years. Acceleration of our method may be achieved by leveraging techniques such as hashgrid (as in InstantNGP [2] / Neuralangelo [3]) and tensor representation (as in TensoRF [4]), and it would definitely be an interesting avenue of future work.
>
>
>
> > The thin shape examples in Figure 4 and Figure 7 don't have complicated geometry. If there is a more complicated geometry, how well would the proposed method perform in reconstructing / modeling? For instance, when dropping a cloth onto an object, the cloth will have a lot of folds, wrinkles and even a lot of self-contacts. Can the proposed method deal with this case?
>
>
> We have included an example of a shape with a more complex surface in the appendix. For clothing with more wrinkles, as long as the resolution is high enough, the details can be well-represented. To see how well fine-level details can be represented with a grid-based representation, we refer the reviewer to Fig. 13 and Fig. 14 in the Nvdiffrec paper [5] which considers reconstruction tasks for watertight meshes parameterized with tetrahedral grids (of resolution 128, the same as those in most of the experiments in our paper). As in G-Shell a non-watertight mesh is parameterized by cutting watertight meshes, the representation capability of modeling fine-level details is similar.
>
> As self-contact breaks the manifold assumption, almost all methods using SDF to represent shapes are not able to handle such cases, and neither is G-Shell. Still, we would like to emphasize that even for simple open surfaces like plain T-shirts, the previous methods (mostly UDF-based) are very sensitive to noise in either the optimization process or datasets. Indeed, this is the major reason why the UDF-based methods fail to reconstruct meshes from realistically-rendered images (i.e., rendered with global illumination — the light rays can bounce between surfaces multiple times).
>
> ---
>
> [1] NeRF: Representing Scenes as Neural Radiance Fields for View Synthesis. Ben Mildenhall, Pratul P. Srinivasan, Matthew Tancik, Jonathan T. Barron, Ravi Ramamoorthi, Ren Ng. ECCV 2020.
>
> [2] Instant Neural Graphics Primitives with a Multiresolution Hash Encoding. Thomas Müller, Alex Evans, Christoph Schied, Alexander Keller. SIGGRAPH 2022.
>
> [3] Neuralangelo: High-Fidelity Neural Surface Reconstruction. Zhaoshuo Li, Thomas Müller, Alex Evans, Russell H. Taylor, Mathias Unberath, Ming-Yu Liu, Chen-Hsuan Lin. CVPR 2023.
>
> [4] TensoRF: Tensorial Radiance Fields. Anpei Chen, Zexiang Xu, Andreas Geiger, Jingyi Yu, Hao Su. ECCV 2022.
>
> [5] Extracting Triangular 3D Models, Materials, and Lighting From Images. Jacob Munkberg, Jon Hasselgren, Tianchang Shen, Jun Gao, Wenzheng Chen, Alex Evans, Thomas Müller, Sanja Fidler. CVPR 2022.

---

### Author Response · Authors · 2023-11-16
**General Response to the Reviewers**

We sincerely thank the reviewers for their time and effort in reviewing our paper and making valuable suggestions. We have uploaded a revised draft of our paper. The major differences are:

- A richer set of qualitative results (for both reconstruction and generation tasks).
- Better visualization of the reconstructed mesh by coloring triangles with back-facing normals orange (the normal directions are made consistent before visualization, with built-in functions in Blender).
- Results on G-Shell w/ FlexiCubes (an isosurface extraction method for watertight meshes that uses a cubic grid instead of a tetrahedral one); this produces better mesh topology.
- Fig. 3 and Appendix E to illustrate our method in a clearer way.
- A polished related work section, with necessary citations suggested by Reviewer hJqu.
- A polished description on the mSDF regularization for reconstruction.
- More results on 1) reconstruction of a shape with a complex pattern on the surface and 2) qualitative results on the effect of mSDF regularization (as suggested by the reviewers Reviewer hJqu and Reviewer dsfR).
- Reconstruction experiment results of our method on real data collected by a hand-held smartphone (per Reviewer 9iXa).
- A more detailed description of MeshDiffusion and some important regularization losses in the appendix.
- Revised visualization of the generalized winding number field (suggested by Reviewer hJqu).
- Visualization for the final watertight mesh template after training (suggested by Reviewer 9iXa).
- Fixed inaccurate descriptions (e.g., generalized winding number, per Reviewer hJqu).

On the latest revision: fixing typos in figure captions in the appendix.

---

### Author Response · Authors · 2023-11-22
**Following up on reviewer's concerns**

Dear Reviewers and AC,

We appreciate again for your time and efforts in reviewing our paper and going through the possibly long responses to your questions. As it is coming to the end of the rebuttal period, we would like to learn if we have addressed your concerns or the reviewers have any other questions. If any, we are more than happy to answer them.

Many Thanks,

Paper 4227 Authors

---

> ### Comment · Reviewer_hJqu · 2023-11-23
> **Thank you**
>
> Thank you for your extensive responses to my questions and for the explanatory revisions as well as extended experiments. I continue to recommend accepting this paper.

---

### Meta-Review · Area_Chair_ahCA · 2023-12-06

**Metareview:**

This paper presents a 3D grid representation with an additional manifold signed distance field field defined on grid vertices for modeling open surfaces, name G-Shell. The proposed representation enables reconstruction from multiview images and generative modeling of both watertight and non-watertight meshes of arbitrary topology. The experiment showcases the effectiveness of the proposed representation in non-watertight mesh reconstruction and generation while retaining the effectiveness for watertight meshes.

The major strengths of the paper are:
(1) Excellent idea of the new representation. It is a differentiable and efficient implicit representation for both watertight and non-watertight meshes.
(2) The results clearly demonstrate the effectiveness of the proposed representation.
(3) High originality and significant contribution.

On the other hand, the weaknesses are:
(1) Limited real-world experiments.
(2) Computational demand is high.

Overall, reviewers and AC are positive about the paper. While the weaknesses above were pointed out in the review and they were not fully addressed, the merit of the paper overwhelms these drawbacks. The reviewers and AC read the rebuttal and took it into consideration to reach the final recommendation.

**Justification For Why Not Higher Score:**

N/A

**Justification For Why Not Lower Score:**

Although some negative concerns were expressed during the review, e.g., high-computation cost and limited real-world experiments, they were addressed by the authors' rebuttal. The paper clearly showcases the merit of its unique contribution.

---

### Decision · Program_Chairs · 2024-01-16

Accept (oral)